# PPAR-γ in Melanoma and Immune Cells: Insights into Disease Pathogenesis and Therapeutic Implications

**DOI:** 10.3390/cells14070534

**Published:** 2025-04-02

**Authors:** Vladimir Sobolev, Ekaterina Tchepourina, Anna Soboleva, Elena Denisova, Irina Korsunskaya, Alexandre Mezentsev

**Affiliations:** 1Laboratory of Physicochemical and Genetic Problems in Dermatology, Center for Theoretical Problems of Physico-Chemical Pharmacology, Russian Academy of Sciences, Moscow 109029, Russia; vlsobolew@gmail.com (V.S.); tchepourina@mail.ru (E.T.); annasobo@mail.ru (A.S.); edenissova@rambler.ru (E.D.); marykor@bk.ru (I.K.); 2Moscow Center of Dermatovenerology and Cosmetology, Moscow 119071, Russia

**Keywords:** PPAR-γ, melanoma, nevus, dysplastic nevus, pigmentation, proliferation, differentiation, apoptosis, tumor microenvironment

## Abstract

Changes in skin pigmentation, like hyperpigmentation or moles, can affect appearance and social life. Unlike locally containable moles, malignant melanomas are aggressive and can spread rapidly, disproportionately affecting younger individuals with a high potential for metastasis. Research has shown that the peroxisome proliferator-activated receptor gamma (PPAR-γ) and its ligands exhibit protective effects against melanoma. As a transcription factor, PPAR-γ is crucial in functions like fatty acid storage and glucose metabolism. Activation of PPAR-γ promotes lipid uptake and enhances sensitivity to insulin. In many cases, it also inhibits the growth of cancer cell lines, like breast, gastric, lung, and prostate cancer. In melanoma, PPAR-γ regulates cell proliferation, differentiation, apoptosis, and survival. During tumorigenesis, it controls metabolic changes and the immunogenicity of stromal cells. PPAR-γ agonists can target hypoxia-induced angiogenesis in tumor therapy, but their effects on tumors can be suppressive or promotional, depending on the tumor environment. Published data show that PPAR-γ-targeting agents can be effective in specific groups of patients, but further studies are needed to understand lesser-known biological effects of PPAR-γ and address the existing safety concerns. This review provides a summary of the current understanding of PPAR-γ and its involvement in melanoma.

## 1. Hypo- and Hyperpigmentation of the Skin, Nevi and Melanoma

Changes in skin pigmentation are one of the major concerns for many people due to the impact on their appearance and social life. The common changes include two main categories: hyperpigmentation and hypopigmentation. Hyperpigmentation refers to the darkening of certain areas of the skin due to an increased production of melanin, manifesting in various forms such as freckles (ephelides), melasma and chloasma, sun-induced age spots (solar lentigines), and post-inflammatory hyperpigmentation [1,2]. In turn, hypopigmentation, which is a reduced biosynthesis of melanin, typically refers to vitiligo, albinism, and tinea versicolor [3]. Since changes in skin pigmentation significantly influence people’s emotional well-being and overall quality of their life, their accurate diagnosis and treatment are crucial to alleviate distress and prevent complications.

Variations in the pigmentation of the skin depend on several factors: both environmental and genetic with key contributors including exposure to UV radiation, hormonal changes, and skin injuries. For instance, skin nevi, commonly referred to as moles (Figure 1), develop from clusters of melanocytes—the specialized skin cells responsible for producing melanin [4]. Compared to regular melanocytes, nevus cells are larger, have fewer dendrites (the branching structures typical of melanocytes), and have a more abundant cytoplasm. Like melanocytes, nevus cells contain coarse granules filled with melanin and share melanin with keratinocytes. However, the accumulation of melanin makes nevus cells more pigmented than regular melanocytes [5].

Since nevi are technically benign, nevus cells divide more frequently than regular skin cells [6]. Primarily, nevus cells localize in the basal layer of the epidermis where they form a small cluster also known as a nest. Then, the cells gradually migrate upward toward the skin surface. As these cells ascend, they contribute to increasing pigmentation, marking the visible location of the nevus. The pigmentation intensity, texture, and size of the nevus vary based on genetic factors, the cellular microenvironment, hormones, and exposure to UV light, creating a contrast with the surrounding skin [7]

Over time, nevus cells may migrate deeper into the dermis, transitioning from junctional nevi (Figure 2a) to compound and dermal nevi (Figure 2b and Figure 2c, respectively), characterized by a more abundant presence of nevus cells in the dermal layer [8]. As these cells move deeper, their visibility in the upper layers of the epidermis diminishes, often resulting in a fading or complete disappearance of pigmentation. In many cases, the boundaries of dermal nevi can become apparent as areas of verrucous skin, creating a textured appearance that distinguishes them from surrounding tissue. This dynamic process highlights the complex behavior of nevus cells and their role in skin pigmentation changes over time.

Nevi exhibit significant variation in appearance, presenting as either flat or raised lesions in a wide range of sizes and shapes. Some nevi are congenital, meaning they are present at birth or develop shortly thereafter, whereas acquired nevi appear later in life, often due to exposure to sun [9]. Among the acquired types, dysplastic nevi exhibit irregular borders, varied colors (ranging from pink to dark brown), and a flat or slightly raised surface [10]. These nevi typically feature a darker central area surrounded by lighter pigmentation, resembling a “fried egg”.

Due to concerns about melanoma, healthcare providers often biopsy dysplastic nevi to confirm their nature and assess any potential malignancy [11]. The likelihood of developing dysplastic nevi over a lifetime depends on a combination of genetic and environmental factors. Genetic predispositions include sensitivity to UV radiation, a pale complexion, red hair, and a tendency to freckle. Environmental factors encompass a history of sunburns, cumulative exposure to the sun, and geographic location, particularly in areas with a high UV index [12].

Unlike a nevus, the progression of melanoma involves two critical phases: the radial growth phase and the vertical growth phase (Figure 3). The radial growth phase includes melanoma in situ, where atypical melanocytes proliferate within the epidermis without invading deeper tissues. This stage may also feature microinvasion into the papillary dermis; however, it lacks significant mitotic activity, indicating a lower potential for metastasis [13]. During this phase, melanoma typically presents as a flat or slightly elevated skin lesion. On physical examination, it appears as irregular plaques with varying pigmentation, including shades of brown, black, blue, or pink, depending on the level of melanin deposited in the cells [14,15]. The radial growth phase may last for an extended period—sometimes for years—before transitioning to the vertical growth phase [16].

The transition to the vertical growth phase marks a pivotal point in melanoma progression, as the tumor acquires malignant properties that enable it to invade surrounding tissues and metastasize rapidly to distant organs. Histologically, the vertical growth phase looks like a community of cohesive nests, clusters, and nodules of tumor cells that are larger and more pleomorphic than those found in the intraepidermal component. This phase often exhibits mitotic activity, indicating aggressive behavior [17]. Clinically, vertical-growth-phase melanoma presents as a pigmented or amelanotic nodule that arises on top of a pre-existing macule or plaque. In some instances, such as with nodular melanoma, it can emerge abruptly without undergoing the radial growth phase [17,18]. The presence of vertical growth signifies an increased risk for metastasis, with studies showing that early vertical-growth-phase melanoma can have a metastasis risk of approximately 10% within eight years of follow-up.

The aggressive nature of melanoma is underscored by its capacity for metastasis, which is often facilitated by factors such as ulceration and specific histological characteristics of the tumor [19]. Unlike nevus cells, which remain confined within their tissue microenvironment, melanoma cells can invade surrounding tissues, distinguishing them despite a higher proliferation rate compared to normal melanocytes. These differences highlight the importance of understanding cellular characteristics when evaluating benign versus malignant skin lesions. Traditional diagnostic criteria for melanoma, such as tumor thickness and mitotic rate, can be insufficient for accurately predicting whether a nevus will transform into melanoma and the speed of its progression [20]. Consequently, there is a growing demand for innovative therapeutic strategies that improve the accuracy and efficiency of melanoma diagnosis and treatment. In particular, the development of combination therapies that integrate approaches with complementary mechanisms of action—such as immunotherapy and small-molecule inhibitors with well-documented anticancer effects—holds significant promise. These strategies aim to enhance treatment efficacy by targeting multiple pathways involved in melanoma progression, overcoming resistance, and improving patient outcomes [21].

## 2. The Expression and Regulation of PPAR-γ

Peroxisome proliferator-activated receptor γ (PPAR-γ/NR1C3) is one of three PPARs encoded in the human genome. Each PPAR has unique tissue distributions and distinct functions [22,23]. The expression of PPAR-α (NR1C1) occurs in tissues involved in the oxidation of fatty acids, such as the liver and muscle, where it plays a crucial role in lipid metabolism and energy homeostasis [24,25]. The expression of PPAR-β/δ (NR1C2) is ubiquitous since this protein regulates the metabolism of fatty acids, expenditure of energy, and inflammation, although its specific functions are less well understood compared to the other subtypes [26]. Participating in multiple signaling pathways (Table 1), PPAR-γ predominantly engaged in the cellular digestion of lipids via anabolic processes regulates the differentiation of adipocytes and sensitivity to insulin [27,28]. Moreover, the activation of PPAR-γ reduces the rate of cell proliferation and modulates the inflammatory response [28,29,30].

In human skin, the expression of PPAR-γ is evident in epidermal keratinocytes [43], melanocytes [44], sebocytes [45], and fibroblasts [46]. PPAR-γ is detectable in the epidermis, the inner root sheath of hair follicles (cytoplasmatic staining), sebaceous glands, and adipocytes of the subcutaneous fat tissue [47]. A gradual increase in the expression of PPAR-γ from basal to apical cell layers occurs in epidermal keratinocytes, where stimulation of PPAR-γ promotes differentiation in vitro [16].

The normal human melanocytes express all three PPAR subtypes: PPAR-α, PPAR-β/δ, and PPAR-γ [43]. The agonists of PPAR-α (WY-146430), and PPAR-γ (ciglitazone/CGZ) inhibit the proliferation of melanocytes in a dose-dependent manner. Moreover, they cause a shape change in the melanocytes, increasing the number of dendrites and the number of keranocytes that they serve compared to untreated cells [48]. In addition, both agonists increase the activity of the DOPA enzyme and pigmentation in human melanocytes by 20–40% [43].

In this section, we discuss how cells regulate the transcriptional activity of PPAR-γ, including ligand-dependent (Section 2.1, Section 2.2 and Section 2.3) and ligand-independent (Section 2.4) mechanisms. We analyze the specificity and mode of action of extensively studied agonists (Section 2.1) and antagonists (Section 2.2), highlighting their therapeutic potential and limitations, as demonstrated by clinical and experimental studies. Furthermore, we emphasize the role of coactivators (Section 2.5) and corepressors (Section 2.6) in modulating the expression of PPAR-γ target genes through recruitment and chromatin remodeling, highlighting the importance of these transcriptional regulators in fine-tuning the biological activities of PPAR-γ. By analyzing these different regulatory mechanisms, we provide a comprehensive understanding of how the transcriptional activity of PPAR-γ is precisely controlled and how these regulatory mechanisms collectively influence its biological outcomes.

### 2.1. The Ligand-Dependent Activation of PPAR-γ

Similar to other members of the nuclear receptor superfamily, PPAR-γ functions as a ligand-activated transcription factor that regulates gene expression. The ligands of PPAR-γ, including both agonists and antagonists (Section 2.2), play a pivotal role in tissue-specific modulation of the receptor. Activation of PPAR-γ can occur either endogenously or exogenously through polyunsaturated fatty acids, fatty acid derivatives, such as 15-deoxy-Δ12,14-prostaglandin J2 (15d-PGJ2), and nitrated fatty acids [49], or through thiazolidinediones (TZDs) and novel synthetic compounds with non-TZD scaffolds, such as F12016 [50], VSP-51 [51], or VSP-17 [52]—Figure 4. The anticarcinogenic potential of PPAR-γ agonists lies in their ability to inhibit tumor growth by interfering with critical signaling pathways or enhancing tumor sensitivity to therapy. This dual mechanism highlights their potential for use in combination with other therapeutic agents, offering promising avenues for cancer treatment.

#### 2.1.1. Thiazolidinediones

TZDs (Figure 4), also known as glitazones, are among the most well-known synthetic ligands of PPAR-γ. They gained significant attention as insulin sensitizers due to their ability to promote fatty acid storage in adipocytes, thereby prioritizing carbohydrate oxidation —particularly glucose—for cellular energy supply in type 2 diabetes (T2D). In preclinical studies, TZDs have demonstrated anticancer properties. These include mechanisms such as cell cycle arrest, apoptosis, hormonal modulation, stromal regulation, and partial depletion of intracellular calcium stores [53,54]. The promising experimental data led to the initiation of multiple clinical studies for different types of cancer, including melanoma (Table 2).

At the same time, the anticancer effects of TZDs are not entirely dependent on their ability to activate PPAR-γ. This conclusion is evident due to multiple studies (e.g., [54,61,62]). First, their effectiveness in killing cancer cells is more related to their chemical structure rather than their ability to activate PPAR-γ. Second, cancer cells with low levels of PPAR-γ can respond more effectively to TZDs than those with high levels. Third, modified versions of TZDs that do not activate PPAR-γ can still effectively induce cancer cell death. Additionally, reducing PPAR-γ levels does not necessarily diminish the ability of TZDs to promote apoptosis [63].

#### 2.1.2. Ciglitazone

CGZ is the prototype of TZDs but has never received approval for clinical use in treating T2D due to its limited therapeutic efficacy [63]. However, it played a significant role in the development and marketing of more potent insulin sensitizers, such as pioglitazone (PGZ), rosiglitazone (RGZ), and troglitazone (TGZ). Later, CGZ regained attention as a novel potential antitumor agent, attributed to its ability to reduce cell proliferation rates and influence the transcriptional activity and expression of PPAR-γ at various differentiation stages of tumor cells. However, many of the antitumor effects exhibited by CGZ appear to be independent of PPAR-γ activation [54,63].

Specifically, CGZ suppresses tumorigenesis by acting as a cyclin D1-ablative agent through proteasomal degradation, leading to cell cycle arrest during the transition from the G_1_ to S phase of the cell cycle [64]. It also promotes calcium efflux from intracellular stores, which results in the deactivation of eukaryotic initiation factor 2 (eIF2), thereby inhibiting the initiation step of translation [62]. At higher concentrations, CGZ activates apoptotic pathways in cancer cells [54,63]. For instance, CGZ has potential as a chemopreventative agent due to its ability to promote the differentiation of cancer cells [65,66].

#### 2.1.3. Troglitazone

TGZ is a dual agonist of PPAR-α and PPAR-γ, with a stronger inhibitory effect on the latter [63]. It was the first antidiabetic medication to show promise in reducing the risk of cardiovascular events associated with T2D [67,68]. At the transcriptional level, TGZ exerts its effects through the transrepression and transactivation of genes that modulate the inflammatory response and regulate the proliferation, migration, and differentiation of cancer cells [65,69]. However, TGZ is currently not in clinical use due to its association with severe liver failure and fatalities resulting from hepatotoxicity.

Previously published data suggest that TGZ exerts its anticancer effects primarily through direct interaction with the epidermal growth factor receptor (EGFR), rather than through binding and activation of PPARs. Upon interacting with EGFR, TGZ facilitates the internalization of the receptor and its subsequent degradation in lysosomes [70,71]. This process reduces the sensitivity of tumor cells to epidermal growth factor (EGF), decreases EGF-dependent phosphorylation of PKB, and ultimately inhibits the growth of tumor cells [71]. In addition, TGZ induces mitochondria-mediated apoptosis and modestly suppresses the migration of cancer cells by modulating the c-Jun N-terminal kinase (JNK) pathway [72].

#### 2.1.4. Pioglitazone

PGZ represents the second generation of TZDs that received approval from the FDA for T2D in 1999. PGZ inhibits the growth and invasion of glioma cells and reduces the abundance of β-catenin in a PPAR-γ-dependent manner in vitro and in xenografts [73]. Treatment with PGZ also inhibits the proliferation and migration of breast cancer cells and tumor growth in PPAR-γ-overexpressing mice. These biological effects relied on the activation of PPAR-γ by PGZ and coincided with phosphorylation of JAK2 and STAT3 [74].

To date, several phase II clinical trials have assessed the efficacy of PGZ as an adjuvant agent in treating patients with various types of cancer. In a double-blind, randomized, placebo-controlled phase II clinical trial for lung cancer, researchers evaluated its efficacy in high-risk current or former smokers with sputum cytologic atypia or known endobronchial dysplasia. The results [75] indicated that PGZ could be effective in subjects meeting the entry criteria (current or former smokers with ≥10 pack-year smoking history, at least mild sputum cytologic atypia, airflow limitation, predicted forced expiratory volume < 0.70, or a history of biopsy-proven endobronchial dysplasia). Another randomized phase II clinical trial evaluated PGZ in combination with chemotherapy for metastatic breast cancer. Investigators assessed whether adding PGZ to a chemotherapy regimen could improve treatment responses. Their findings [76] suggested that the combination therapy resulted in higher complete radiologic response rates and stable disease status compared to chemotherapy alone.

The only phase I clinical trial [56] involving melanoma patients explored the safety of PGZ in combination with other agents, including the DNA alkylating agent trofosfamide and the inhibitor of cycloxygenase 2 (PTGS2/COX2) rofecoxib. The authors aimed to analyze the safety of this proposed treatment regimen. The study demonstrated that patients tolerated the treatment well, with no major side effects observed. Notably, the treatment stabilized the disease in 11% of participants for 6 months, resulted in complete remission in one patient, and extended progression-free survival by 2.8 months in patients with chemorefractory melanoma. Given that some patients achieved partial and complete remissions, the authors expressed optimism about promoting apoptosis in melanoma cells and increasing their susceptibility to chemotherapy through the use of PPAR-γ agonists in conjunction with a PTGS2 inhibitor and trofosfamide. However, the subsequent withdrawal of rofecoxib in 2004 hindered their ability to conduct further studies.

#### 2.1.5. Rosiglitazone

RGZ received approval from the Food and Drug Administration (FDA) as an insulin sensitizer for T2D in 1999. However, the FDA restricted its usage after reviewing the results of a meta-analysis linking RGZ to a higher risk of cardiovascular events [77]. However, those restrictions were lifted after the FDA reviewed the results of a trial failing to show the alleged risk of heart attack. Similar to other TZDs, some anticancer effects of RGZ may depend on the activation of PPAR-γ. For instance, the pro-apoptotic effect of RGZ in the hepatocarcinoma cell line HepG2 and esophageal cancer cells (EC109 and TE10) clearly depended on the activation of PPAR-γ and was reduced in the presence of antagonists [78,79].

The activation of PPAR-γ is also necessary to enforce the chemoprotective effect of RGZ on cancer cells. According to Cao et al. [80], RGZ improves the sensitivity of hepatocellular carcinoma (HCC) cell lines BEL-7402 and Huh-7 to the chemostatic agent fluorouracil, 5-FU. This effect depended on the activation of PPAR-γ, which upregulates phosphatase and tensin homolog deleted on chromosome 10 (PTEN) and downregulates PTGS2. Following the activation, the former inhibits the PI3K/PKB (phosphoinositide 3-kinase/protein kinase B) signaling pathway, whereas the latter contributes to the synergistic effects of 5-FU. At the same time, the effect of RGZ on cell proliferation can be either positive or negative. For instance, it reduced the growth of cultured non-small cell lung cancer (NSCLC) cell lines H1792 and H1838 in both PPAR-γ-dependent or -independent manners. Specifically, the antiproliferative effect of RGZ was sensitive to the presence of PPAR-γ antagonist GW9662. However, it was reversible by overexpression of the PPAR-γ2 isoform [81].

RGZ modulates the immune-suppressive effects that cancer cells exert on tumor-associated immune cells, although their connection to PPAR-γ remains unknown. According to Konger et al. [82], treatment with RGZ (8 mg/kg/day) reduces the number of myeloid-derived suppressor cells (MDSCs) associated with the tumor microenvironment (TME). By decreasing these immunosuppressive cells, RGZ helps restore the activity of effector T cells, thereby enhancing the overall antitumor response in xenograft animal models of non-melanoma skin cancer. Moreover, RGZ promotes the infiltration of TME by CD8^+^ and CD4^+^ subsets of CD3^+^ T cells (CD—complex of differentiation), which are essential for immune-mediated tumor rejection. RGZ is also capable of producing potent anti-angiogenic effects. In a xenograft model of adrenocortical carcinoma, 5 mg/kg/day RZD significantly reduced the expression of angiogenic vascular endothelial growth factor (VEGF) and a key marker for vascular endothelial cells, CD31, compared to untreated controls [83]. Finally, RGZ is capable of inhibiting insulin-like growth factor 1 (IGF1) or hypoxia-inducible factor 1α (HIF1α) in myeloma cells [84].

At the molecular level, one PPAR-γ-dependent effect of RGZ is the induction of *PTEN.* It occurs due to the binding of RGZ-activated PPAR-γ to the promoter of *PTEN*, which prevents the activation of PKB in HCC cell lines, by primarily targeting the PI3K/PKB signaling pathway [85]. Moreover, RGZ promotes the expression of the pro-apoptotic genes breast cancer gene 1 (*BRCA1*) in MCF-7 cells [86,87], and BCL2-associated X protein (*BAX*) in human pancreatic carcinoma cells [88]. Among the signaling pathways, the RGZ-activated PPAR-γ inhibits tumor growth via interference with the phosphorylation of extracellular signal-regulated kinases 1/2 (ERK1/2) [89]. Moreover, it improves drug sensitivity by suppressing multidrug resistance 1 (MDR1/P-gp) in melanoma cells [90].

Despite the studies assessing the clinical value of RGZ, where this agonist of PPAR-γ previously demonstrated promising results, there are concerns about using RGZ as monotherapy due to its ability to activate paracrine signaling in metastatic melanoma cells [91]. Several phase I clinical trials for cancer concluded that RGZ is safe to use as an adjuvant for chemotherapeutic drugs [55,92,93]. For instance, a pilot study, conducted by Yee et al. [93], demonstrated that RGZ was well tolerated without serious adverse events. The authors reported that RGZ increased the sensitivity to insulin and decreased the level of insulin in patients’ blood. It also increased the level of adiponectin in blood serum. However, it did not reduce the proliferation of tumor cells.

Some clinical trials for RGZ completed phase II [94,95]. A study on liposarcoma [95] did not identify significant changes in the histologic appearance of the liposarcomas by the treatment despite previous reports that RGZ is capable of terminally differentiating cultured liposarcoma cells [96]. The changes in gene expression did not transform into a clinical response. Moreover, increased activity of PPAR-γ did not correlate with the clinical evolution of the tumors. Consequently, the authors concluded that RGZ was not effective as an antitumor drug in the treatment of liposarcomas. In a study on RGZ for thyroglobulin-positive and radioiodine-negative differentiated thyroid cancer [94], the researchers did not identify clinically important toxicity associated with RGZ. They mentioned positive dynamics associated with the disease flow on radioiodine scans, although half the patients did not respond to the therapy. Despite this, the therapeutic effect did not correlate with the expression of PPAR-γ at either the mRNA or protein levels in the responders.

#### 2.1.6. Efatutazone

Efatutazone (EFZ)/Inolitazone is a next-generation TZD drug that functions as a selective agonist of PPAR-γ. In studies utilizing NSCLC cell lines derived from PC-9, specifically the EGFR-TKI-resistant variants PC-9ER and PC-9ZD, EFZ effectively inhibits cell motility, primarily by antagonizing the TGF-β/SMAD2 (transforming growth factor-β/mothers against decapentaplegic homolog 2) signaling pathway [97]. However, EFZ does not exhibit significant growth-inhibitory effects on these resistant cells, suggesting its potential utility as an adjuvant therapy in combination with other agents, such as erlotinib or T0901317 [97,98].

When administered at low doses over a short duration, EFZ enhances the differentiation of lactational and luminal cells in mouse models of breast cancer [99]. This effect follows the activation of PPAR-γ, which subsequently reduces the phosphorylation of PKB while leaving signaling through the ERK pathway unaffected. The authors suggest that EFZ could be beneficial for human patients with cancer due to its ability to delay the progression of breast cancer in mice, by promoting differentiation over proliferation.

Clinical trials of EFZ have demonstrated that it is well tolerated, with manageable side effects [100,101]. Results from these studies indicated an acceptable toxicity profile and evidence of disease control in treated patients. Notably, a phase I trial combining EFZ with paclitaxel for anaplastic thyroid cancer showed promising outcomes: nearly half of the participants (7 out of 15) achieved stable disease [101]. The median time to disease progression also increased in a dose-dependent manner. Specifically, patients receiving 0.15 mg of EFZ had a median progression time of 48 days, while those on 0.3 mg experienced a median progression time of 68 days. Additionally, the corresponding median survival times were 98 and 138 days, respectively. Moreover, two phase II clinical trials were initiated to investigate the therapeutic effects of EFZ in myxoid liposarcoma (ClinicalTrials.gov ID: NCT02249949) and anaplastic thyroid cancer (ClinicalTrials.gov ID: NCT02152137). However, EFZ was discontinued due to slow enrollment and limited efficacy as a single agent [102].

#### 2.1.7. Endogenous and Natural Agonists of PPAR-γ

The endogenous ligands of PPAR-γ constitute the J-series of prostaglandins and arachidonic acids (e.g., 15d-PGJ2), other long-chain polyunsaturated fatty acids (LC-PUFA), eicosanoids, nitroalkanes, and oxidized phospholipids including docosahexanoic acid (DHA) [103]. Acting in a PPAR-γ-dependent and -independent manner, they inhibit the growth of cancer cells by inducing their apoptosis and reducing cancer-related angiogenesis [104]. For instance, 15d-PGJ2 inhibits the nuclear factor κB (NFκB) signaling pathway by forming covalent bonds with the IκB kinase complex and the p65 subunit of NFκB. These interactions interfere with the activation of NFκB and subsequent expression of pro-inflammatory cytokines, like TNF-α and IL6 (tumor necrosis factor α and interleukin 6, respectively) [105].

Moreover, many bioactive compounds derived from plants may potentiate the transcriptional activity of PPAR-γ by acting as its natural ligands [103] (Figure 5). Some of them, like tocopherols (α and γ), increased the expression of *PTEN* and downregulated *PKB* in a rat esophageal cancer model [106]. A noteworthy example of these natural products is bavachinin, a bioactive compound isolated from seeds of *Psoralea corylifolia*, which suppresses the growth of A549 lung cancer cells by promoting the activation of PPAR-γ and generation of ROS [107]. The natural ligand of PPAR-γ, known as resveratrol, exhibits antiproliferative effects on uterine sarcoma cells by blocking the wingless-related integration site (WNT)/β-catenin pathway in a PPAR-γ-dependent manner [108]. Acting on the same signaling pathway, another agonist of PPAR-γ, curcumin, induces G_2_ cell cycle arrest and apoptosis in diffuse large B-cell lymphoma [109].

According to Flori et al. [110], the interaction of PPAR-γ with 2,4,6-Octatrienoic acid (Octa) enhances cellular defense against oxidative stress induced by UV light. In epidermal keratinocytes, this interaction upregulates two key antioxidant enzymes, heme oxygenase-1 (HMOX1) and catalase (CAT), through the binding of the retinoic X receptor α (RXR-α)–PPAR-γ heterodimer to peroxisome proliferator response elements (PPREs) in their promoters. Octa also exhibits a protective effect against UVB-induced damage, by activating DNA repair mechanisms for photodamage and double-strand breaks. Cells with activated PPAR-γ display higher viability rates compared to controls. Moreover, Octa reduces the number of sunburn cells and double-strand breaks in DNA in both pre- and post-UVB-irradiated skin equivalents. These results are consistent with previous findings by Wang et al. [111], who reported that mice lacking RXR-α, the heterodimerization partner of PPAR-γ, exhibited higher apoptosis rates, impaired epidermal proliferation, and increased DNA damage in epidermal keratinocytes when exposed to UVB than mice with normal levels of the receptor.

Natural and endogenous agonists of PPAR-γ have emerged as promising candidates in cancer research due to their dual role in modulating tumor biology and inflammatory pathways. Unlike TDZs, these compounds, such as curcumin, resveratrol, and PUFAs, are generally non-toxic and prioritize patient safety, leveraging their anti-inflammatory and anticancer effects [112,113]. Their therapeutic potential has driven investigations in clinical trials for melanoma (Table 2) and other malignancies, including liver, colon, and breast cancers, where they influence cancer progression by inducing apoptosis and cell cycle arrest in tumor cells, upregulating differentiation markers to promote less aggressive phenotypes, and suppressing pro-inflammatory cytokines (e.g., TNF-α, IL6) and tumor neovascularization [114,115]. At the same time, many of these compounds face challenges, such as limited bioavailability, partial agonism, and underrepresentation in conventional clinical trials.

### 2.2. Selective Antagonists of PPAR-γ

Unlike PPAR-γ agonists, antagonists inhibit the activation of the receptor by blocking the recruitment of coactivators and facilitating the interactions of PPAR-γ with corepressors. This, in turn, downregulates PPAR-γ target genes, resulting in distinct biological effects. Clinically, antagonists of PPAR-γ (Figure 6) have been shown to enhance glucose uptake, making them potentially beneficial in managing metabolic disorders, such as T2D. Depending on the experimental conditions, these antagonists can stimulate osteogenesis and inhibit adipogenesis, offering potential therapeutic applications in conditions like osteoporosis and obesity (reviewed in [28,116]). Furthermore, specific PPAR-γ antagonists have demonstrated potent anticancer activity in preclinical studies, establishing a promising reputation in this area.

#### 2.2.1. MM902

The computational search for a molecular target of the novel anticancer agent MM902 (3-(3-(bromomethyl)-5-(4-(tert-butyl) phenyl)-1H-1,2,4-triazol-1-yl) phenol) surprisingly identified this compound as a specific antagonist of PPAR-γ [117] with ~10 times less affinity to either PPAR-α or PPAR-β/δ. Additionally, MM902 showed moderate binding activity with a mutant form of BRAF (V600E) and IKK-α (BRAF—B-rapidly accelerated fibrosarcoma; IKK-α—inhibitor of nuclear factor κB kinase subunit α). According to previous studies, both identified proteins had a strong association with malignant melanoma. The BRAF mutation V600E is present in approximately 50% of malignant melanomas [118,119]. In turn, the activation of the NFκB signaling pathway is also frequent in melanoma cells [119]. By forming covalent bonds within the ligand-binding domain of PPAR-γ, MM902 effectively prevents the interaction of agonists with the receptor. According to the authors of [117], this experimental compound demonstrates potent efficacy in inhibiting the growth of a wide range of cancer cells, including leukemia, melanoma, lung, colon, central nervous system (CNS), ovarian, kidney, prostate, and breast cancers, with a GI_50_ value of less than 1 μM.

#### 2.2.2. T0070907

T0070907 is a potent and selective antagonist of PPAR-γ, functioning by covalently modifying a Cys285 on the receptor [120]. This modification blocks the necessary conformational changes in the structure of PPAR-γ, thereby inhibiting its activation [121]. Similar to MM902, T0070907 prevents the recruitment of coactivators to PPAR-γ, instead promoting the recruitment of corepressors. Studies using mutant receptors have demonstrated that T0070907 modulates the interaction between PPAR-γ and cofactor proteins by affecting the conformation of the ligand-binding domain (LBD). As a result, modification of the receptor with T0070907 repressed the transcription of PPAR-γ target genes. Notably, inactivation of PPAR-γ by T0070907 did not prevent its interactions with its binding partner RXR-α, even after the activation of RXR-α by its specific agonist LGD1069 [121].

At the same time, T0070907 exhibits potent antitumorigenic activity, inducing G_2_/M phase arrest in cervical cancer cells and enhancing the effects of radiation therapy, thereby emerging as a potential radiosensitizer [122]. Furthermore, T0070907 disrupts microtubule dynamics during cell division, by reducing tubulin abundance in cancer cells. In tested cell lines, this disruption leads to cell cycle arrest and apoptosis [121,122]. Notably, T0070907 also reduces the invasive potential and motility of cervical and breast cancer cells in a PPAR-γ-independent manner [122,123]. However, T0070907 has not been pursued in clinical trials despite promising results in experimental biology.

#### 2.2.3. GW9662

GW9662 is one of the most frequently used antagonists of PPAR-γ. It covalently modifies the Cys285 residue in LBD of PPAR-γ, thereby preventing the binding and activation of the receptor by agonists [120,124]. Although GW9662 effectively blocks agonist-mediated activation, it exhibits negligible effects on the basal transcriptional activity of PPAR-γ. This distinguishes it from antagonists like T0070907, which actively repress basal transcription. The structural differences between T0070907 and GW9662 and PPAR-γ explain the weaker interaction between GW9662 and PPAR-γ. Unlike T0070907, GW9662 lacks an acceptor of hydrogen bond necessary for interaction with Arg288, a key residue that stabilizes PPAR-γ in a corepressor-selective conformation. As a result, GW9662 primarily functions as a neutral antagonist, whereas T0070907 acts as an inverse agonist by repressing transcription [120].

According to the previous studies, GW9662 inhibits the growth of breast cancer cells MCF7 and MDA-MB-231 [125]. The authors showed that GW9662 also enhanced the growth-inhibitory effects of PPAR-γ agonists, like RGZ, suggesting the presence of potential PPAR-γ-independent pathways that would contribute to its anticancer activity. In addition to its effects on proliferation of cancer cells, GW9662 has demonstrated promising benefits for metabolic health, such as improved insulin sensitivity and anti-inflammatory properties. Using GW9662 in combination with anti-PD-L1 (programmed death ligand 1) in a murine model of melanoma significantly improved the effectiveness of immunotherapy in female hosts [126,127]. Currently, GW9662 is primarily utilized in research settings to investigate PPAR-γ-mediated signaling pathways and their implications in metabolic disorders and cancer biology. Its ability to inhibit the interaction of PPAR-γ with agonists has established it as an important tool for studying this process. However, there is no available information regarding completed or ongoing clinical trials involving GW9662.

### 2.3. PPAR-γ Ligands as a Therapeutic Strategy

Collectively, these findings indicate that PPAR-γ ligands offer promising therapeutic strategies for anticancer therapy, particularly in melanoma, by enhancing immune responses and disrupting tumor growth. Due to their success in preclinical studies, several PPAR-γ agonists have progressed to clinical trials, demonstrating high specificity for their primary target compared to other PPARs, as well as a broad spectrum of PPAR-γ-independent anticancer effects (Table 3). Specifically, PPAR-γ ligands have shown the ability to inhibit tumor growth, induce terminal differentiation of cancer cells, cause cell cycle arrest, and trigger apoptosis, highlighting their potential as anticancer agents. Furthermore, as shown below, modulation by the transcriptional activity of PPAR-y would be a potential approach to disintegrate the TME, rendering it more susceptible to immune surveillance and attack, which further enhances the anticancer effects. The combination of PPAR-γ ligands with other therapies, such as chemotherapy and immunotherapy, is anticipated to boost their efficacy and provide a more comprehensive treatment approach, potentially mitigating existing adverse effects. Overall, these findings highlight the trend towards developing novel PPAR-γ ligands, including both agonists and antagonists, with improved pharmacological profiles, reduced limitations, and milder side effects, aiming to fully exploit their therapeutic potential and offer more effective and targeted treatment options for melanoma and other cancers.

### 2.4. Ligand-Independent Changes in the Transcriptional Activity of PPAR-γ

In addition to its canonical ligand-dependent transcriptional activity, PPAR-γ retains activation capacity under ligand-depleted conditions through its ligand-independent activation domain [172]. In this case, the activation follows the post-transcriptional modifications of the receptor by acetylation (Figure 7). Increased acetylation of PPAR-γ results in its activation even in the absence of external ligands, leading to an acceleration of lipid catabolism. The acetylation of PPAR-γ and its ligand-independent activation also follow the exposure of cells to PGZ [173]. Although the physiological relevance of this phenomenon remains understudied, its role in lipid biosynthesis pathways highlights therapeutic potential for cancer, particularly in targeting lipid metabolism in cancer cells [174]. Furthermore, acetylation of PPAR-γ contributes to the activation of macrophages—a critical component of both host antitumor defense and the TME, where macrophages paradoxically enable immune evasion [175].

Phosphorylation of specific serine residues (Ser114 and Ser273; Figure 7) represents another critical post-translational modification influencing the transcriptional activity of PPAR-γ [176]. The cyclin-dependent kinase 5 (CDK5)-mediated phosphorylation of Ser273 in adipose tissue disrupts the normal transcriptional function of PPAR-γ without impairing its ligand-binding capacity [177]. This modification selectively alters the expression of a subset of PPAR-γ target genes and shifts the recruitment of cofactors, favoring interactions with corepressors over coactivators [177]. Inhibition of the phosphorylation by SR1664 restores the transcriptional activity of PPAR-γ without inducing full agonism [178]. The previous studies have shown that cancer cells exposed to DNA-damaging agents such as carboplatin, etoposide, and doxorubicin exhibit a higher level of phosphorylated Ser273. Blocking this phosphorylation—either genetically or pharmacologically—sensitizes cancer cells to apoptosis, enhancing the efficacy of chemotherapy [179,180].

The phosphorylation of Ser114 or Ser112 in the murine PPAR-γ by MAPKs (ERK or JNK) suppresses the transcriptional activity of PPAR, reducing the expression of its target genes [41,42]. However, the substitution of Ser82 by Ala (corresponds to Ser112 in the murine γ1-isoform) prevents the PDGF-mediated repression of PPAR-γ in mice and cultured human cells that inhibits both ligand-dependent and -independent transactivating functions [42,181] and affects the ability of PPAR-γ to recruit coactivators [182]. Conversely, Cdk9-mediated phosphorylation of Ser112 in adipocytes exerts opposing effects, enhancing the transcriptional activity of PPAR-γ [183]. This dichotomy highlights that phosphorylation outcomes depend on the cellular context, introducing complexity in the regulation of PPAR-γ-dependent genes.

The transcriptional activity of PPAR-γ is modulated by various post-translational modifications, including SUMOylation and O-GlcNAcylation. As demonstrated by Shimizu et al., SUMOylation of Lys107 negatively regulates the transcription of PPAR-γ target genes by repressing the ligand-independent transactivation function of the A/B domain, thereby reducing the overall transactivating competence of PPAR-γ [184]. In contrast, the substitution of Lys107 with Arg enhances the transcriptional activity of PPAR-γ [185]. Similarly, β-O-linked N-acetylglucosamine (O-GlcNAc) modification at Thr54 in PPAR-γ1, located within the same A/B domain, also suppresses its transactivatory function [186].

### 2.5. Coactivators of PPAR-γ

Conformational changes induced by agonists in PPAR-γ are essential for regulation of its transcriptional activity (Figure 8). When an agonist binds to the LBD of a receptor, it initiates a cascade of structural changes. These changes promote the recruitment of coactivators to the receptor and also facilitate the release of corepressors. Furthermore, they are critical for the formation of functional heterodimers, which are indispensable in the regulation of gene expression.

PPAR-γ coactivator-1α (PGC-1α) is probably the most extensively studied coactivator of PPAR-γ. In melanocytes, the induction of PGC-1α increases after the stimulation of cells by α-melanocyte-stimulating hormone (α-MSH). Acting upstream of protein kinase A (PKA), α-MSH also stabilizes PGC-1α at the protein level. In turn, PGC-1α and its homologue protein PGC-1β positively regulate the expression of *MITF*, *TRPM1*, and *TYR* (*MITF*—microphthalmia-associated transcription factor; *TRPM1*—transient receptor potential melastatin 1; *TYR*—tyrosinase) [187,188]. Consequentially, the overexpression of PGC-1α stimulates pigmentation in human melanocytes, human and mouse melanoma cells, and TetO::PGC-1α transgenic animals (TetO—tetracycline operator) [187,189].

Knocking PGC-1α down significantly diminishes the previously mentioned biological effects. Additionally, silencing PGC-1β reduces PGC-1α expression, indicating that PGC-1β regulates PGC-1α levels. There is also a reciprocal influence of MITF and PGC-1α on the expression of each other [187]. Consequently, it is not surprising that several single-nucleotide polymorphisms (SNPs) in the PGC-1β gene (*PPARGC1B*) demonstrate an evident association with the ability to tan [190]. Notably, one specific SNP (rs32579) is more frequent in individuals with higher expression of *PPARGC1B*. Carriers of this SNP also have a 20% lower risk of melanoma. In contrast, there is no such association known for SNPs in *PPARGC1A* [187]. In addition to its role with PPAR-γ, PGC-1α and its homolog PGC-1β also interact with other transcription factors, including other nuclear receptors [191]. Moreover, PGC-1α indirectly activates the negative regulator of melanogenesis and transcription factor NRF2, by inhibiting GSK-3β (NRF2—nuclear factor erythroid 2-related factor 2; GSK-3β—glycogen synthase kinase-3β) [192,193].

In melanoma, a higher expression of PGC-1α indicates a lower survival rate [189]. The heterogeneous expression of PGC-1α within tumors contributes to variations in tumor behavior. Tumor cells exhibiting reduced PGC-1α activity tend to be pro-metastatic. Contrarily, those with higher PGC-1α levels demonstrate significantly higher proliferation rates. This heterogeneity in the expression of PGC-1α is critical for the progression of melanoma, allowing it to respond to different signals, switching between survival–proliferation and invasion–metastasis [194].

Docking of PGC-1α to PPAR-γ induces further conformational changes in the receptor, facilitating subsequent interactions between PPAR-γ and NCoA1/nuclear receptor coactivators (NCoA1, -2 and -3). Their binding to PPAR-γ leads to recruitment of CREB-binding protein (CBP) and p300. By possessing histone acetyltransferase activity, these proteins modify chromatin by adding acetyl groups to histones. The modifications of histones promote the transition of chromatin to the open configuration, making the PPAR-γ target genes more accessible to the transcriptional machinery [195].

The abundance of CBP, p300, NCoA1, and NCoA3 is significantly higher in malignant melanoma, where they promote tumor cell progression through the cell cycle [196,197,198,199]. CBP, p300, and NCoA3 also enhance the viability of melanoma cells by activating the DNA damage responses. Specifically, NCoA3 inactivates the tumor suppressors checkpoint kinase 2 (CHK2), tumor protein p53 (TP53), and p21 [199]. In turn, CBP and p300 activate proteins involved in base excision repair, nucleotide excision repair, and non-homologous end joining [200]. Moreover, at least two coactivators of PPAR-γ, NCoA1 and p300, serve as prognostic factors for melanoma. Specifically, higher expression of NCoA3 predicts a worse outcome [201]. In contrast, a positive outcome correlates with higher nuclear localization of p300 and a negative outcome with its accumulation in the cytoplasm [202].

### 2.6. Corepressors of PPAR-γ

Another group of proteins that modulate the transcriptional activity of PPAR-γ includes corepressors, specifically the nuclear receptor corepressor (NCoR) and the silencing mediator of retinoid and thyroid hormone receptors (SMRTs). These corepressors typically interact with the unliganded form of PPAR-γ within the complex with RXR-α [203,204]. Notably, certain antagonists of PPAR-γ, such as T0070907 [120], can stabilize it in a repressive conformation upon binding. This interaction enhances the association of PPAR-γ with corepressors, which, in turn, inhibits the recruitment of coactivators [205]. As a result, complexes formed between PPAR-γ and corepressors lack transcriptional activity.

Upon binding with an agonist, PPAR-γ undergoes a conformational change that leads to the release of corepressors and facilitates the recruitment of coactivators. This transition is crucial for shifting PPAR-γ from a repressive to an active transcriptional state [206,207]. Although PPAR-γ is present in both the cytoplasm and the nucleus [28], its interactions with corepressors, such as NCoR and SMRT, predominantly occur in the nucleus [35,206,207]. In the cytoplasm, PPAR-γ typically does not engage with these corepressors. Instead, it interacts with heat shock proteins and RXR-α until it shuttles to the nucleus, regardless of ligand binding [208]. Once in the nucleus, PPAR-γ forms inhibitory complexes with corepressors that recruit HDACs [206,207]. These HDACs play a crucial role in transitioning chromatin to the closed state and repressing transcription by deacetylating histones. Specifically, when HDACs are part of an inhibitory complex with PPAR-γ and RXR-α, they effectively suppress the expression of PPAR-γ target genes.

Both NCoR and SMRT play critical roles in melanoma progression and hold significant potential as prognostic markers. In benign melanocytic nevi, both corepressors exhibit strong nuclear localization. However, their biological effects on melanoma cells extend beyond the repression of PPAR-γ, as they interact with a wide range of transcription factors. This broader role suggests that the role of their interaction with PPAR-γ on melanoma biology requires further investigation.

For instance, both healthy and benign melanocytes predominantly display the nuclear localization of NCoR. In contrast, many melanoma cells accumulate NCoR in the cytoplasm. According to Gallardo et al. [209], the cytoplasmic localization of NCoR is a negative prognostic factor for melanoma patients, correlating with worse clinical outcomes. At the molecular level, this mislocalization coincides with the activation of the NF-κB signaling pathway, which is regulated by many factors, including PPAR-γ. At the same time, the inhibition of the IKK produces the opposite effect. However, there is currently no evidence to suggest that activation of PPAR-γ could lead to translocation of NCoR back to the nucleus.

## 3. Activation of PPAR-γ in Melanoma Cells and Its Biological Effects

### 3.1. Activation of PPAR-γ in Melanoma Cells by Stimulation of Melanocortin 1 Receptor

The abundance of PPAR-γ in melanoma cells is typically higher compared to normal melanocytes [210]. It is also evident in benign nevi [47] because of higher demands on energy in proliferating cells [127]. On the other hand, the expression of PPAR-γ poorly correlates with protein levels, suggesting that melanoma cells prefer to stabilize PPAR-γ rather than activate the transcription of the PPAR-γ-encoding gene. Among patient-resected melanoma and various melanoma cell lines, it varies in a high range due to their high clonal heterogenicity [91]. Studies have also shown that PPAR-γ localizes predominantly in the cytoplasm, suggesting a balance shift between its ligand-free and ligand-bound forms in favor of the unliganded one [47,127].

The melanocortin-1 receptor (MC1R) is one of five melanocortin receptors encoded in the human genome. These receptors are seven-transmembrane G-protein-coupled receptors whose activation stimulates the production of cyclic adenosine monophosphate (cAMP) [211,212]. MC1R is the principal receptor responsible for melanogenesis, which includes the biosynthesis of eumelanin and pheomelanin (black and red melanin) [212]. The prevalence of eumelanin in the skin cells preserves the integrity of their DNA and makes it more resistant to damage produced by UV light [213]. The primary downstream signaling pathway (Figure 9) relies on the production of cAMP by adenylate cyclase, leading to the phosphorylation (activation) of CREB and the induction of genes directly participating in melanogenesis, such as *TYR*, tyrosinase-related protein-1 (*TRP1*), and tyrosinase-related protein-2 (*TRP2*) [214]. However, single-nucleotide polymorphisms (SNPs) in the MC1R gene may impair its ability to activate cAMP, shifting the balance between eumelanin and pheomelanin in favor of the latter. This reduces the protective effect and increases susceptibility to skin cancers [211,212]. In addition to the cAMP pathway, wild-type MC1R activates two signaling pathways, MAPK-ERK and PKB, further enhancing its own functionality (MAPK—mitogen-activated protein kinase) [212].

As a master regulator of melanogenesis, MC1R interacts with other signaling pathways, including PPAR-γ. Given the potent anticancer effects of natural and synthetic PPAR-γ ligands, the interplay between MC1R and PPAR-γ presents an attractive avenue for developing novel anti-melanoma therapies. Studies indicate that stimulation of MC1R in murine melanoma cells by its natural ligand α-MSH activates PPAR-γ [215]. This activation leads to the translocation of PPAR-γ to the nucleus and initiates the transcription of PPAR-γ target genes [216]. Mechanistically, these events follow the activation of phospholipase C (PLC) [215] and calcium efflux from intracellular stores to the cytoplasm [217]. Then, the released calcium ions activate PLA2 by hydrolizing phospholipids. Phospholipase A2 (PLA2) produces arachidonic acid and its derivatives from triglycerides, which are natural inducers of PPARs. Moreover, the stimulation of MC1R increases the expression of PPAR-γ although most of it accumulates in the cytoplasm unless stimulated with agonists [215,218]. At the same time, the activation of PPAR-γ does not require the activation of the canonical MC1R signaling pathway because it does not depend on the intracellular level of cAMP [215]. However, receptors homologous to MC1R, such as MC3R-MC5R, can also be a part of anticancer effects exhibited by PPAR-γ [219,220].

### 3.2. The Role of PPAR-γ in Pigmentation

For patients with advanced melanoma, hyperpigmentation of melanoma lesions is one of the recognized negative prognostic factors [221]. It correlates with resistance to radiotherapy [222,223] and reduces the efficacy of chemotherapy. When therapeutic agents enter melanosomes within melanoma cells, they covalently interact with melanin, and become trapped and unable to reach their primary target—genomic DNA [224,225]. Then, the cell excretes the neutralized molecules into the extracellular matrix [224].

This mechanism effectively shields melanoma cells from various chemotherapeutic agents, including DNA intercalating agents and topoisomerase inhibitors (e.g., cisplatin, etoposide, daunorubicin, and doxorubicin), as well as cytoskeleton-disrupting toxins, like vinblastine, irrespective of their mode of action [225,226]. Given this resistance mechanism, the downregulation of melanogenesis represents a promising strategy to enhance the sensitivity of melanoma to cytotoxic drugs [224]. Targeting transcription factors capable of modulating melanogenesis could therefore serve as a potential molecular approach for anticancer therapy [227].

The contribution of PPAR-γ and its agonists to the regulation of melanogenesis is well documented. The agonists of PPAR-γ (e.g., WY14643, Octa, CGZ) stimulate melanogenesis by increasing the activity of tyrosinase without changing its expression level [44,48]. Moreover, WY14643 and CGZ increase the transcriptional activity of *MITF* [48]. At the same time, human patients with T2D treated with a TZD do not show any alteration in skin pigmentation, suggesting that these drugs do not alter functionality of normal melanocytes [228].

Due to the changes in gene expression and cell metabolism, the outcomes of melanogenesis for melanoma cells treated with PPAR-γ agonists can be opposite. Specifically, a higher resistance of hyperpigmented cells to xenobiotics requires a higher concentration of drugs to produce comparable biological effects with an increasing risk of off-target effects. For instance, Octa and RGZ reduce melanogenesis and tyrosinase activity in mouse melanoma cells (B16F1) [229]. Depigmentation also occurs after overexpression of PPAR-α in the same cell line [230] or treatment with the PPAR-α agonist fenofibrate. Contrarily, WY14643 and CGZ produce the opposite effect in B16F1 [230] and human melanoma A375 cells, respectively. Moreover, CGZ evidently induces MITF that, in turn, induces the tyrosinase-encoding gene, *TYR* [231].

Similar to PPAR-γ, the other nuclear receptors also contribute to melanogenesis. For instance, the biological effects produced by agonists of the vitamin D receptor (VDR) are comparable to those produced by TZDs [232,233]. In turn, liver X receptors (LXRs) have been implicated in melanogenesis by modulating MITF degradation through ERK-mediated pathways, thereby affecting TYR activity and melanin production [234]. Collectively, these findings suggest that the ability of nuclear receptors to influence pigmentation and tumorigenesis underscores their therapeutic potential in conditions such as melanoma, where dysregulated melanogenesis contributes to tumor progression. Future studies targeting these receptors may provide new insights into their roles in pigmentation disorders and cancer therapy.

### 3.3. Effect of PPAR-γ on Proliferation

The previous studies have shown that treatment of various melanoma cell lines with agonists of PPAR-γ inhibited proliferation in a dose-dependent manner [47,210,231], although high concentrations (>10 μM) were necessary to achieve sufficient changes in cell proliferation rates. On the other hand, the reduction in cell proliferation rate was significant in the cells at early and advanced stages of melanomagenesis [231]. Some agonists of PPAR-γ (e.g., PGZ and CGZ) also demonstrated an evident antiproliferative effect in vivo [64,235]. Moreover, they inhibited tumor growth in animal models. For instance, both PGZ and CGZ reduced tumor size [236] and CGZ also negatively affected the progression of human melanoma xenografts in nude mice [64].

At the molecular level, the antiproliferative effects shown by the agonists can be either dependent on or independent of PPAR-γ. For instance, the agonists modulate the NF-κB signaling pathway in a PPAR-γ-independent manner. In melanoma cells, the NF-κB signaling pathway contributes to the deregulation of the cell cycle by influencing the expression of proteins critical for the transition through the G_1_/S checkpoint, such as cyclin D1 (CCND1) and CDK2 [237]. By overexpressing CCND1 and CDK2, melanoma cells manage to bypass the mechanisms controlling the cell cycle and increase their proliferation rate. In turn, PPAR-γ inhibits the activation of NF-κB by upregulating IκBα, a negative regulator of NF-κB [33]. Moreover, it acts as an E3 ubiquitin ligase delivering Lys48-linked polyubiquitin to the NF-κB subunit p65, marking it for degradation [34]. They also cause G_1_/S cell cycle arrest [236].

### 3.4. Role of PPAR-γ in Initiation of Apoptosis

PPARs, particularly PPAR-γ, do not cause apoptosis in healthy melanocytes [235]. However, their ligands play a role in the initiation of apoptosis in melanoma cells. Specifically, the agonists of PPAR-γ (natural or synthetic) induce apoptosis in a variety of cancer cells including lymphoma, multiple myeloma, bladder, gastric, esophageal, pancreatic, hepatoma, colon, breast, brain, and lung cancer cells (rev. in [238]). The induction of apoptosis in cancer cells by agonists involves both PPARγ-dependent and -independent mechanisms, with their efficacy and specificity being modulated by the concentration of the agonist and the genetic background of the cancer cells.

The TZDs (CGZ, PGZ, etc.) reduce cell viability in a dose-dependent manner and can induce apoptosis in melanoma cells [235]. Relatively low concentrations of agonists, like CGZ (0.1–1 μM), lock the cells in G_1_/S arrest, significantly decreasing their proliferation rate. The mechanism underlying cell cycle arrest includes changes in gene expression, such as upregulation of p21 (*CDKN1A*) and hypophosphorylation of retinoblastoma 1 (RB1/pRB) [64]. Moreover, CGZ downregulates cyclin D1 [239] and promotes its degradation in proteasomes [240]. However, these changes are not enough to induce apoptosis [64,231]. In contrast, higher concentrations of CGZ (>10 μM) can be cytotoxic, leading to cell blebbing, detachment, and positive staining for annexin V, which indicates apoptosis in cultured melanoma cells and in vivo [231,235]. Specifically, 10–15 μM CGZ reduces the viability of the melanoma cell lines WM35, A375, and 501 mel by more than 30% in a time-dependent manner [231].

The molecular mechanisms underlying TDZ-induced apoptosis can vary across different types of cancer. According to Plissonnier et al. [241], treatment with 30 μM CGZ induced apoptosis in cervical cancer cells (HeLa, Ca Ski, C-33 A) through the activation of the death receptor signaling pathway. Specifically, treatment with CGZ promoted apoptosis by upregulating the expression of death receptors DR4 and DR5, as well as TNF-related apoptosis-inducing ligand (TRAIL), in the affected cancer cells. Conversely, it downregulated the levels of cellular flice-inhibitory protein (c-FLIP), a negative regulator of apoptosis, and eIF2, a positive regulator of translation. Furthermore, treatment with CGZ triggered a calcium efflux from the endoplasmic reticulum, leading to the activation of the protein kinases PERK (protein kinase RNA-like endoplasmic reticulum kinase) and PKR (protein kinase RNA-activated). These kinases, in turn, inhibited the translation of mRNA, by phosphorylating eIF2α, thereby contributing to the induction of apoptosis.

Comparing CGZ to other PPAR-γ agonists, such as RGZ and 15d-PGJ2, Plissonnier et al. [241] found that they shared this mechanism and all acted independently of PPAR-γ. This conclusion is consistent with previous studies, which demonstrated that RGZ (300 μM) increased DR5 expression in Caki renal cancer cells through the generation of ROS in a dose- and time-dependent manner [242]. Similarly, 15d-PGJ2, the final metabolite produced by dehydration of prostaglandin D2, stabilized DR5 mRNA in PC3 human prostate cancer cells [243]. Furthermore, a study by Weng et al. showed that structural homologues of CGZ and TGZ, which are unable to interact with PPAR-γ, still affected the viability of melanoma cells [159].

The antiproliferative effect of PPAR-γ agonists, such as CGZ and other TZDs, has been observed in clinical trials involving patients with liposarcoma, prostate, or breast cancer [244,245,246]. These trials provided evidence supporting the role of PPAR-γ in inhibiting the growth of tumor cells. However, further studies of TZDs have been suspended despite this potential benefit due to concerns of hepatotoxicity and other adverse effects, highlighting the critical need to balance therapeutic efficacy with patient safety [247]. The development of novel PPAR-γ modulators with improved safety profiles or alternative strategies to target the PPAR-γ pathway may offer a way to harness its anticancer potential without the associated risks [248].

### 3.5. Influence of PPAR-γ on Terminal Differentiation of Melanoma Cells

In the complex landscape of cellular differentiation, PPAR-γ emerges as a pivotal regulator, influencing cell fate through diverse mechanisms encompassing transcriptional control and metabolic modulation. A primary function of PPAR-γ is to induce the expression of genes essential for cellular differentiation. For example, in preadipocytes, activation of PPAR-γ by agonists directly upregulates the genes of lipid metabolism and storage, thereby promoting the maturation of preadipocytes into adipocytes. Similarly, causing an accumulation of lipids in breast cancer cells, the agonists of PPAR-γ help them to acquire a more differentiated state [249]. The affected cells reduce their growth rate and clonogenic capacity. Moreover, the agonists of PPAR-γ promote the terminal differentiation in several cancer cell lines of different origins including colon [250], colorectal [251,252], lung [253,254], and prostate [255,256] cancer. For instance, TGZ induces the differentiation of cultured human colon cancer cells by upregulating the expression of biomarkers characteristic of differentiated enterocytes, namely villin 1 (*VILL1*) and alkaline phosphatase (*ALPI*) [250].

Treatment of normal melanocytes with TZDs induces marked morphological alterations [48]. Specifically, melanocytes exhibit increased dendricity and cytoplasmic enlargement, accompanied by an extended melanin-supplying network relative to non-stimulated cells. This phenotypic transition correlates with elevated expression of melanocyte-specific markers, including TYR, TRP1, and TRP2. It also leads to a concomitant enhancement in the production of melanin. These TZD-treated melanocytes also display increased pigmentation and enhanced maturation of melanosomes. The activation of PPAR-γ by TZDs is thought to mediate these effects, as it modulates the expression of genes implicated in both melanocyte differentiation and melanin biosynthesis, ultimately resulting in an augmented melanin-supplying capacity of the melanocytes.

The pro-differentiation effects of PPAR-γ agonists (e.g., CGZ, RGZ, and 15d-PGJ2) on cultured melanoma cells are well documented [64,89,257]. For instance, Liu et al. [89] mentioned that RGZ induces the terminal differentiation of melanoma cell lines within several days. Specifically, they showed that treatment of A375 cells with RGZ for 72 h inhibited their growth. RGZ also promoted the accumulation of melanin and increased the tyrosinase activity, suggesting a shift in cultured cells in favor of a more differentiated state, suggesting that PPAR-γ mimics some of the effects promoted by α-MSH in melanocytes. However, the complexity of TME may diminish or overturn these potentially beneficial effects, favoring tumorigenesis [91,258], and a cohort study suggested that the use of TZDs may be associated with an increased risk of melanoma [259].

### 3.6. Role of PPAR-γ in Modulating Tumor Microenvironment

Lipid metabolism controlled by PPARs plays a crucial role in the survival and proliferation of tumor cells. It is also a factor in reprogramming healthy cells associated with TME to protect cancer cells from elimination by the immune system [260]. The TME or tumor stromal community is a local area created and dominated by the tumor. In general, the TME consists of non-malignant host cells and non-cellular components. The cellular components of TME represent a heterogeneous collection of infiltrating and resident host cells, such as immune and endothelial cells, as well as cancer-associated fibroblasts (CAFs) [261,262]. The latter are cells of different origins (normal fibroblasts, pericytes, smooth muscle cells, mesenchymal stem cells, etc.) reprogrammed by the tumor. The number and kinds of cells that reside in TME vary with the type of cancer and stage. The non-cellular components include extracellular matrix (ECM) proteins, signaling molecules, growth factors, and other high- and low-molecular-weight metabolites [262].

In the stroma, normal host cells typically have tumor-suppressing abilities. During carcinogenesis, the stroma undergoes significant changes, leading to a complex interplay between stromal and tumor cells. The evolved stromal cells can either promote or inhibit tumor growth, in a context-dependent and cell-type-specific manner. This dynamic interaction between stromal and tumor cells synergizes with the metabolic activities of tumor through responding to extracellular signals secreted by the tumor, promoting tumor angiogenesis, and establishing peripheral immune tolerance [261]. As a part of their feedback, the tumor-associated cells shape the tumor, influencing whether the tumor regresses and develops resistance to evade the immune system. Non-malignant cells surrounding the tumor mass send signals to the cancer cells, and some, such as CAFs, build a favorable niche for the cancer to thrive. Immune cells residing within the TME can have both pro- and anti-tumorigenic functions. First, the TME can act as a barrier against the infiltration of tumor immune cells. Second, aberrant expression of PD-L1 on tumor cells impedes antitumor immunity, resulting in evasion of the immune system [261,262].

Lipid metabolism is increasingly recognized as a critical player in cancer development, influencing the growth of tumor cells, immune interactions, and therapeutic resistance. The acceleration of lipid metabolism supplies cancer cells with energy, membrane components, and signal molecules necessary to support a higher proliferation rate, acquire malignancy, and improve their resistance to anticancer therapies [263,264,265]. Moreover, some tumor-originated lipids exhibit immunomodulatory effects. For instance, the uptake of fatty acids produced by melanoma cells causes functional inhibition and iron death in CD8^+^ CTLs [266,267]. In addition, some lipids contribute to tumorigenesis, as intercellular signaling molecules [268,269]. In addition, ligand-activated PPAR-γ may exhibit anti-angiogenic effects in vascular endothelial cells, by inhibiting their association into tube-like structures in vitro and choroidal neovascularization in vivo [270].

Being settled in the TME, one of the most important subsets of CTLs, which normally destroy cancer cells by secreting granzymes, has to increase the catabolism of fatty acids due to hypoxia and glucolytic deprivation. Typically, CTLs rely on aerobic glycolysis for energy production and function. However, their exposure to free fatty acids shifts their metabolism towards increasing the oxidation of fatty acids, which does not necessarily support the activation and proliferation of T cells. This metabolic shift also reduces their mitochondrial function, diminishing their ability to respond to tumors [260,271].

Although the role of PPAR-γ in reprogramming CD8^+^ cells still needs further clarification, previous experimental studies suggest that PPAR-γ is a potential factor modulating their cytotoxic effects. Notably, a study by Chowdhury et al. [272] demonstrated that stabilization of the activatory complexes of PPAR-γ and PGC1α by bezafibrate increased the proliferation of naïve T cells and enhanced the effector function in CTLs. Furthermore, bezafibrate augmented their reactivity against tumors during PD-1 blockade, highlighting the potential therapeutic implications of targeting PPAR-γ in cancer immunotherapy.

In turn, decreased transcriptional activity of PPAR-γ in PPAR-γ-knockout mice correlated with a higher number of effector T cells, which is distinguished by increased antigen-specific proliferation and excessive interferon γ (IFNG) production in response to IL12 [273,274]. Similarly, mice deficient in PPAR-γ in CD8^+^ T cells produced a faster and enhanced activation of effector T cells [275] compared to controls. The PPAR-γ-deficient cells also exhibited a higher mitotic index, increased expression of activation markers (e.g., killer cell lectin-like receptor subfamily G member 1KLRG1, CD44 and CD69, cytotoxic T-lymphocyte antigenCTLA4, programmed cell death protein 1PD1, lymphocyte-activation gene 3LAG3 and T-cell immunoglobulin and mucin domain-3TIM3) and higher production of pro-inflammatory cytokines IFNG and TNF-α. In turn, disabling PPAR-γ in memory T cells enhanced their capacity for differentiation to effector cells, increasing the release of pro-inflammatory cytokines. Moreover, PPAR-γ-deficient memory T cells showed a higher expression of activation markers, such as CD69, PD1, and CTLA4.

Tumor-associated regulatory T cells (Tregs) demonstrate significant metabolic adaptations that distinguish them from conventional Tregs, characterized by enhanced glycolytic activity and increased fatty acid uptake and oxidation, which collectively contribute to their immunosuppressive functions within the TME [276,277]. These changes lead to increased accumulation of triglycerides, which in turn inhibits their biological activities [278]. Notably, Tregs can proliferate in low-oxygen and -glucose conditions, enabling them to support tumor cells in evading immune surveillance by inducing peripheral immune tolerance [279,280]. Specifically, they secrete anti-inflammatory cytokines, such as IL10 and TGF-β, in a PPAR-γ-dependent manner, which further suppresses the activation of effector T cells [280].

PPAR-γ also plays a crucial role in the differentiation of Treg cells, as evidenced by the significant reduction in Tregs in mice deficient in PPAR-γ, whereas activation of PPAR-γ has the opposite effect, promoting their differentiation [281]. Conversely, disabling PPAR-γ leads to increased production of effector T cells, which are found circulating in the blood and residing in mesenteric lymph nodes [282]. The PPAR-γ agonist CGZ has been shown to modulate the balance between subpopulations of these T cells, acting through both PPAR-γ-dependent and -independent mechanisms [282,283]. This suggests that modulating PPAR-γ transcriptional activity could be a viable option to disrupt the integrity of the TME and prevent cancer cells from evading the immune system [284].

The interaction between tumor-associated macrophages (TAMs) and the TME is critical in shaping the immune response to cancer, and recent research has emphasized the key role of metabolic changes in influencing macrophage behavior. To sustain their metabolic needs, tumor-associated M2 macrophages primarily rely on the enhancement of oxidative phosphorylation and the β-oxidation of fatty acids [285,286]. The TME is rich in fatty acids, which facilitate the upregulation and activation of PPAR-γ in M2 TAMs, leading to the increased expression of genes encoding fatty acid transporters [287,288]. In contrast, monocytes are more likely to differentiate into pro-inflammatory M1 macrophages when they do not have transporters of fatty acids [289]. The activation of PPAR-γ plays a critical role in the differentiation of newly arrived monocytes towards the M2 phenotype. Upon stimulation with fatty acids and cytokines, such as IL4, PPAR-γ senses the presence of fatty acids and induces the expression of genes associated with the M2 phenotype [290,291].

Another factor that contributes to differentiation of monocytes toward M2 macrophages is the proteolytic inactivation of PPAR-γ by caspase 1 (CASP1) in some cancer cells [32]. In these cells, the activation of CASP1 follows the stimulation of the cells by pro-inflammatory cytokines IL1β and IL18, which in turn promotes its expression through a positive feedback mechanism [292]. After cleavage by CASP1, a truncated PPAR-γ travels to mitochondria where it directly interacts with medium-chain acyl-CoA dehydrogenase (MCAD) [32]. Their interaction inhibits MCAD activity and, consequently, reduces the oxidation of fatty acids. For this reason, the monocytes start accumulating lipid droplets and differentiate into M2 macrophages. At the same time, either pharmaceutical inhibition of CASP1 or the infusion of bone marrow-derived macrophages genetically engineered to overexpress murine MCAD markedly suppresses the tumor growth.

The TME significantly influences the behavior and functionality of dendritic cells (DCs), shaping their ability to regulate immune responses and interact with other immune cells within the tumor. Various factors, including cytokines produced by tumor cells or other immune cells, influence the differentiation of DCs towards more immunosuppressive phenotypes and alter their antigen-presenting capabilities. Notably, an increased abundance of fatty acids in the TME significantly impairs the ability of tumor-associated DCs to process and present antigens. The accumulation of fatty acids and oxidized lipids in the TME leads to the increased uptake and accumulation of lipid droplets by DCs. This leads to downregulation of major histocompatibility complex class I (MHCI) molecules and a reduction in their representation on the cell surface [293]. In contrast, a decrease in fatty acid biosynthesis has been positively correlated with enhanced immunogenicity of DCs [294,295], suggesting that normalizing lipid metabolism in the TME could potentially delay tumor growth and its progression to advanced stages.

Specifically, PPAR-γ plays an active role in shaping the functional characteristics of monocyte-derived DCs [287]. The activation of PPAR-γ promotes their differentiation [296] and changes the expression of various cell surface receptors [287]. Stimulation of PPAR-γ by agonists significantly impacts lipid metabolism in these cells, leading to the induction of genes involved in the immune response [296]. This suggests that PPAR-γ has a positive role in the activation of monocyte-derived DCs associated with tumors, potentially enhancing their immunogenicity and antitumor functions.

The activation of PPAR-γ also enhances the internalizing activity of monocyte-derived DCs, improving their ability to engulf and process antigens, thereby promoting the adaptive immune response [287]. Moreover, PPAR-γ bidirectionally regulates the expression of CD1 glycoproteins, a class of molecules responsible for presenting self and foreign modified lipids to T cells. Specifically, the activation of PPAR-γ downregulates CD1a and upregulates CD1d. Although both CD1a and –d present lipid antigens to T cells [297], CD1d is specifically involved in antigen presentation to CD8-positive natural killer cells whose functionality declines with progression to advanced stages of cancer [260,271].

By scavenging nutrients from the TME, cancer cells acquire a wide range of biomolecules that are essential for their growth, survival, and the production of energy. The interaction with cancer cells gradually transforms CAFs into the donors of commodities for cancer cells. The upregulation and activation of PPAR-γ in CAFs enable them to undergo the necessary metabolic changes [298,299]. CAFs with high PPAR-γ expression exhibit a senescent phenotype [300] and display increased glycolytic activity, producing L-lactate and other metabolites that are essential for the growth of cancer cells [300,301]. To provide even more nutrients to adjacent cancer cells, they also activate autophagy. The impact of CAFs on tumor growth is further demonstrated by their ability to accelerate the growth of tumor xenografts when co-implanted with breast cancer cells, compared to fibroblasts with normal PPAR-γ expression.

The presence of CAFs in the TME also has a profound impact on the behavior of immune cells, ultimately influencing the outcome of the antitumor immune response [302,303]. First, CAFs secrete metabolites, such as lactate, that alter the metabolic state of tumor-associated immune cells, thereby modulating their antitumor activity. Second, CAFs create physical barriers in the path of immune cells to the tumor, by secreting fibrillar proteins, such as collagens and fibronectins, into the ECM. This remodeling enhances ECM stiffness and protects tumor cells from effective immune responses [302]. Third, CAFs produce cytokines, such as IL6 and TGF-β, which limit the functionality of CTLs and DCs, leading to T-cell anergy or dysfunction. Moreover, they secrete chemokines (C-X-C motif chemokine ligand 12—CXCL12/SDF1, IL6, C-C motif chemokine ligand 2—CCL2, etc.), which recruit monocytes and neutrophils to the TME, further contributing to the immunosuppressive environment [302,303].

## 4. Conclusions

As a transcription factor, PPAR-γ regulates diverse biological processes, such as the storage and transport of fatty acids, their metabolism, and the metabolism of glucose [31]. Its activation promotes adipogenesis and lipid uptake in adipocytes, enhancing insulin sensitivity through enhanced sequestration of fatty acids in adipose tissue. Many naturally occurring agents directly bind to and activate PPAR-γ, including PUFAs, such as arachidonic acid and its derivatives, and prostaglandins [28]. Moreover, the agonists of PPAR-γ inhibit proliferation in cultured human breast, gastric, lung, prostate, and other cancer cell lines [304].

Despite the fact that agonists of PPAR-γ exhibit potent anticancer effects, not all of these effects require activation of the receptor. In many cases, the agonists are effective at significantly higher concentrations than typically needed for receptor–ligand interactions. This limitation contributes to their insufficient performance in clinical trials for various types of cancer. Notably, retrospective analyses have indicated an increased risk of developing bladder cancer and melanoma associated with the chronic use of antidiabetic TZDs [259,305]. Supporting this notion, the PPAR-γ agonist RGZ has been shown to promote the growth of tumor cells through the induction of paracrine signaling in metastatic melanoma cells [91]. These mixed outcomes in clinical trials may stem from factors such as patient selection biases (e.g., inclusion of advanced/refractory cases with altered tumor biology) and the complexity of signaling pathways, where activation of PPAR-γ-independent signaling mechanisms confounds therapeutic effects. Additionally, heterogeneity in receptor expression and ligand responsiveness across cancer subtypes contributes to inconsistent results.

This paradoxical duality extends to melanoma, where PPAR-γ produces multiple biological effects. As we have discussed above, PPAR-γ regulates proliferation, differentiation, apoptosis, and survival in melanoma cells. Moreover, it modulates metabolic adaptations and immunogenicity in both melanoma cells and tumor-associated stromal cells [301]. Agonists of PPAR-γ also target hypoxia-induced angiogenesis, offering a potential avenue for anticancer therapy [306]. Controversially, ligands of PPAR-γ paradoxically exhibit either tumor-suppressing or tumor-promoting effects, depending on contextual factors [301]. Their final outcome likely hinges on interactions with components of the TME, such as cancer-associated fibroblasts and endothelial and immune cells, which influence therapeutic balance [301,306]. The current knowledge about PPAR-γ suggests that PPAR-γ-targeting agents can be selectively efficient in certain groups of patients [63].

Taking into consideration the ongoing research, modulation by the biological effects of PPAR-γ can be a beneficial therapeutic approach for cancer patients, particularly patients with melanoma. With the emergence of new PPAR-γ ligands, including both agonists and antagonists, the role of PPAR-γ in clinical practice is expanding significantly. However, our current understanding and experience with PPAR-γ and its ligands remain insufficient to confidently recommend these agents for the treatment of melanoma. Future experimental and clinical studies will hopefully unravel other less-known aspects of these agents and evaluate their efficacy and safety in various dermatological disorders.

## Figures and Tables

**Figure 1 cells-14-00534-f001:**
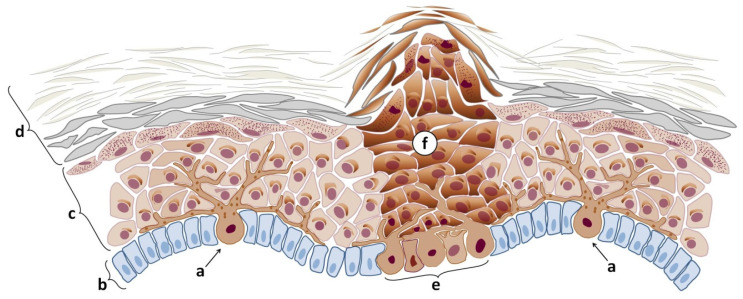
Cellular components of human skin. (a) Melanocytes; (b) basal (non-differentiated) keratinocytes; (c) partially differentiated epidermal keratinocytes of suprabasal, spinal, and granular layers; (d) corneocytes; (e) nevus nest; (f) nevus (mole).

**Figure 2 cells-14-00534-f002:**
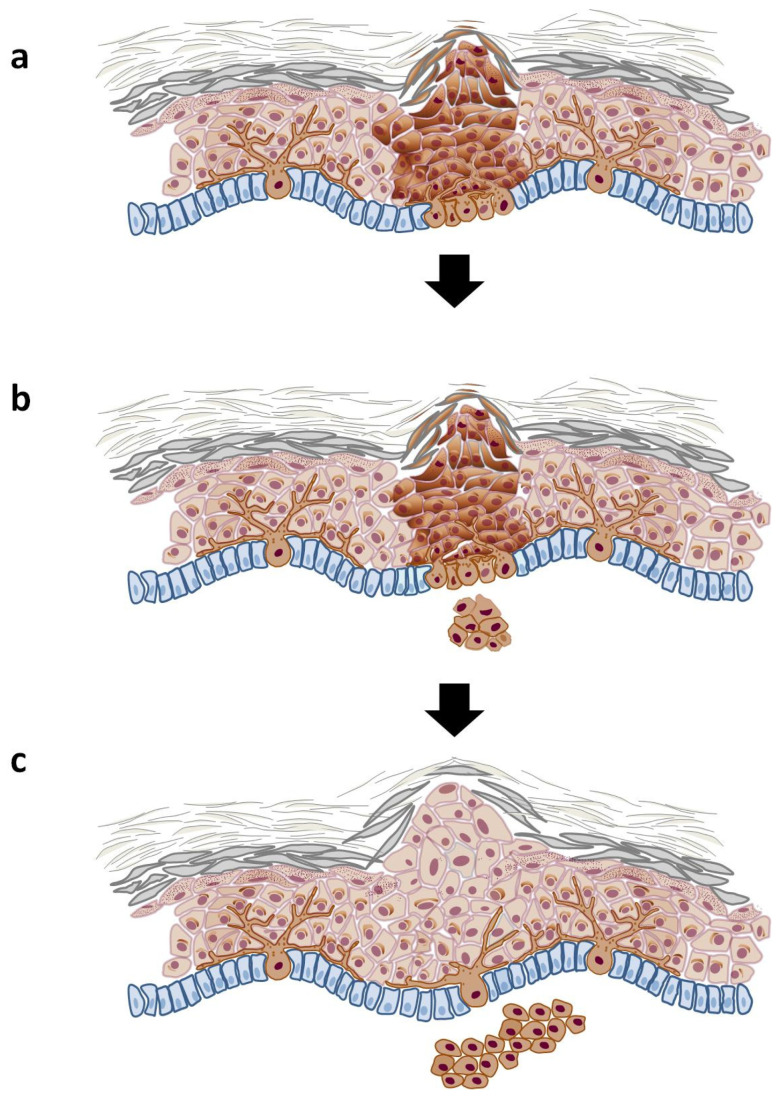
Evolution of melanocytic nevi: transformation from junctional to compound and dermal nevi. (**a**) Junctional nevus: characterized by nests of melanocytes located at the dermo-epidermal junction, with no dermal component. (**b**) Compound nevus: shows both junctional and dermal components, with melanocytic nests at the dermo-epidermal junction and melanocytes extending into the dermis. (**c**) Dermal nevus: features melanocytes exclusively within the dermis, with no junctional component, often resulting in a more raised appearance.

**Figure 3 cells-14-00534-f003:**
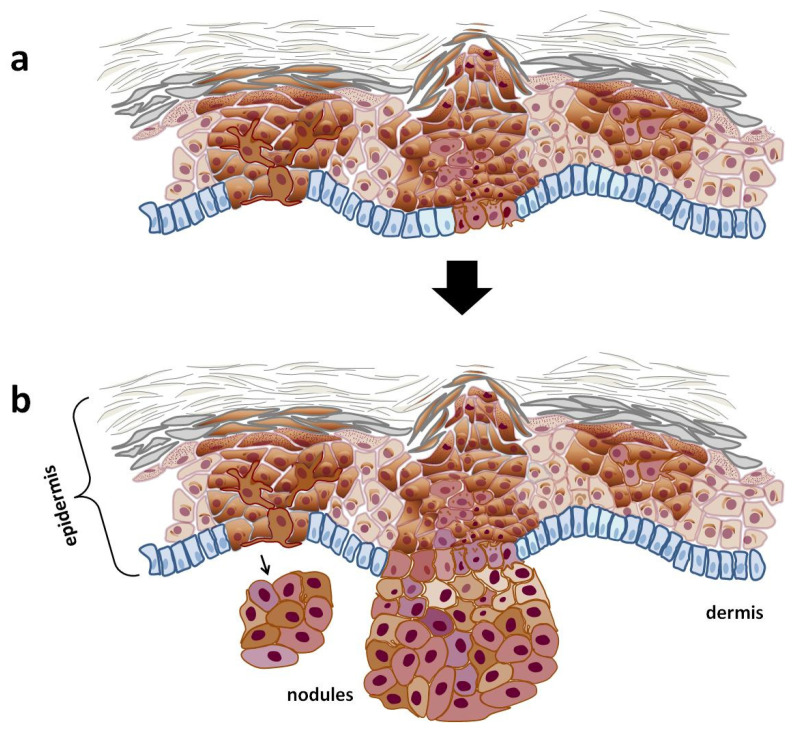
Phases of melanoma growth: radial and vertical expansion. (**a**) Radial growth phase: In this initial stage, melanoma cells proliferate horizontally along the skin surface, often resulting in the formation of irregular plaques. Atypical melanocytes may appear in clusters or as solitary cells throughout the epidermis. This lateral spread of melanoma cells increases the size of the skin lesion without the development of nodules. Mitotic activity is typically minimal at this stage. (**b**) Vertical growth phase: During this stage, melanoma cells penetrate deeper into the dermis, growing perpendicularly to the skin surface and forming distinct nodules or cohesive clusters of tumor cells. In this phase, melanoma cells display increased pleomorphism and a higher frequency of mitotic activity, indicating more aggressive tumor behavior.

**Figure 4 cells-14-00534-f004:**
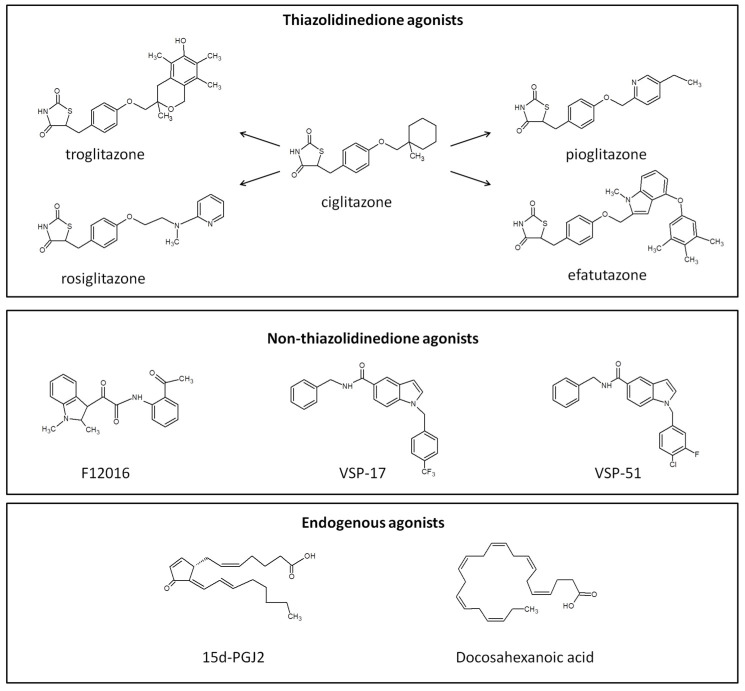
Synthetic and endogenous agonists of PPAR-γ.

**Figure 5 cells-14-00534-f005:**
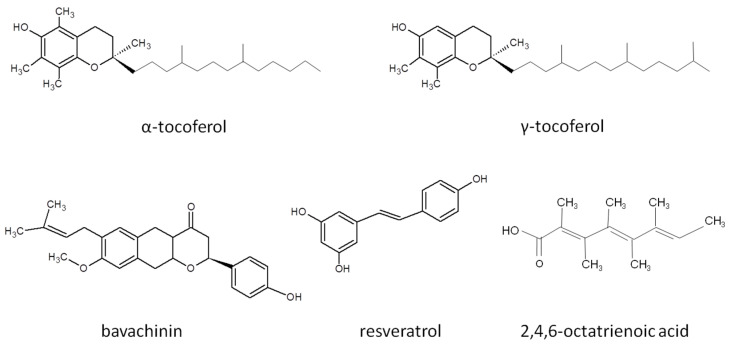
Natural agonists of PPAR-γ.

**Figure 6 cells-14-00534-f006:**
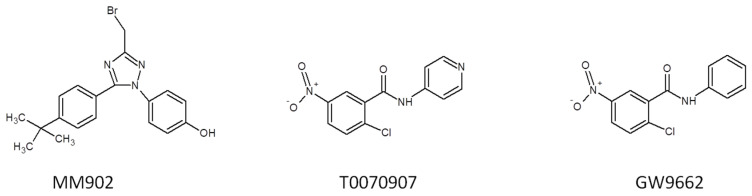
Specific antagonists of PPARγ.

**Figure 7 cells-14-00534-f007:**
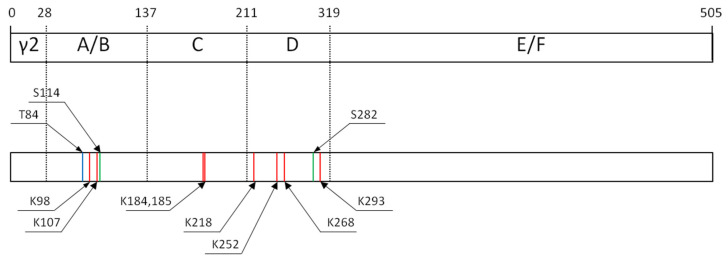
Schematic representation of PPAR-γ structure and post-translational modifications regulating its transcriptional activity. The PPAR-γ molecule comprises distinct domains, including the A/B domain containing the ligand-independent activation function 1 (AF-1), the C domain (DNA-binding domain, DBD) that controls transcriptional activity, ubiquitination, and localization, the D domain that serves as a hinge region, facilitating domain interactions and proper receptor positioning on DNA, and the E/F domain (ligand-binding domain, LBD) containing the ligand-dependent activation function 2 (AF-2). Post-translational modifications, such as lysine acetylation (red lines), serine phosphorylation (green lines), SUMOylation of Lys107, and β-O-linked N-acetylglucosamine (O-GlcNAc) modification (blue line), fine-tune the transcriptional activity of PPAR-γ, influencing its interaction with coactivators and corepressors.

**Figure 8 cells-14-00534-f008:**
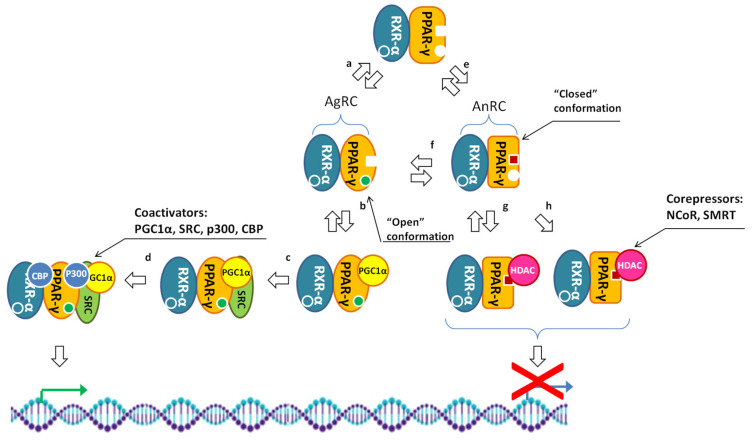
Interaction of PPAR-γ with coactivators and corepressors. PPAR-γ heteromerizes with RXR-α in a ligand-dependent manner (small blue circle inside the RXR-α molecule) in the cell nucleus. Although the preactivation of RXR-α occurs in a PPAR-γ-independent manner, it contributes to the transcriptional activity of the heterodimer. On the other hand, full transcriptional activation typically requires both RXR-α and PPAR-γ to be activated, as these receptors work cooperatively within the heterodimer. (a) As a part of the heterodimer, PPAR-γ interacts with an agonist (small green circle inside the PPAR-γ molecule) to form an agonist–receptor complex (AgRC). This interaction stabilizes PPAR-γ in the “open” conformation, favoring the recruitment of coactivators. Specifically, it creates a docking site for PGC1-α. (b) The binding of PGC1-α leads to additional structural rearrangements in PPAR-γ. (c) When in the nucleus, PPAR-γ recruits NCoA. (d) Then, the protein complex binds to histone acetyltransferases (p300 and CBP), which remodel chromatin and facilitate PPAR-γ binding to PPAR-γ responsive elements (PPREs). (e) Binding to an antagonist (small red square inside the PPAR-γ molecule) leads to the formation of antagonist–receptor complex (AnRC) and stabilizes PPAR-γ in the “closed” conformation. This interaction prevents the recruitment of coactivators to PPAR-γ and facilitates its interaction with corepressors. (f) If the antagonist binding is reversible, PPAR-γ may exchange the antagonist for an agonist and switch back to the “open” conformation. (g) Moreover, PPAR-γ may interact with corepressors (NCoR or SMRT) that exhibit deacetylase activity, remodeling chromatin and preventing DNA-dependent RNA polymerase II from recognizing PPREs and transcribing PPAR-γ target genes. (h) Irreversible modification of the agonist binding site by an antagonist locks PPAR-γ in the “closed” conformation, preventing exchange with an agonist and subsequent activation of PPAR-γ.

**Figure 9 cells-14-00534-f009:**
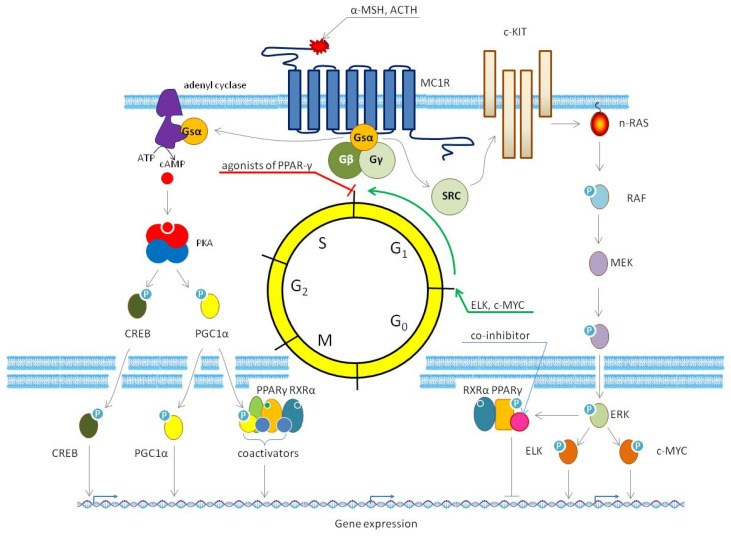
The canonical MC1R signaling pathway. The melanocortin-1 receptor (MC1R), a G-protein-coupled receptor, plays a crucial role in regulating melanogenesis in human skin. Upon binding to its ligands, α-melanocyte-stimulating hormone (α-MSH) or adrenocorticotropic hormone (ACTH), MC1R activates two primary downstream signaling pathways. The cAMP-dependent pathway (**left**) involves the phosphorylation and activation of CREB and PGC1α. The latter is a coactivator of PPAR-γ. The activation of the canonical MC1R signaling pathway transcriptionally regulates the genes involved in melanogenesis, including *TYR*, *TRP1*, *TRP2*, *PMEL*, and *MITF*. Moreover, MC1R activates the MAPK/ERK signaling pathway (**right**), which leads to phosphorylation of PPAR-γ by ERK (specifically, ERK2) and the following decrease in the expression of PPAR-γ-dependent genes. It also promotes cell cycle progression from the G_0_ to G_1_ phase. In addition, the MC1R signaling pathway promotes the terminal differentiation of melanocytes. The coordinated activation of these pathways enables MC1R to regulate melanin production and cell proliferation in melanocytes. In contrast, agonists of PPAR-γ prevent the transition of cells from the G_1_ to G_S_ phase, causing a cell cycle arrest.

**Table 1 cells-14-00534-t001:** PPARγ in intracellular signaling: mechanisms and functional implications.

Pathway	Function of PPAR-γ	Role of PPAR-γ and Physiological Effect	References
The canonical PPAR-γ signaling pathway	PPAR-γ forms a heterodimer with retinoid X receptor α (RXR-α). Upon binding a ligand, this complex recruits either coactivators or corepressors, depending on the ligand type and cellular context. Then, PPAR-γ-RXRα complexes bind to PPAR-responsive elements (PPREs) in DNA, altering gene expression by modifying chromatin structure and recruiting the transcriptional machinery.	In this pathway, PPAR-γ functions as a nuclear receptor and transcription factor. As a central regulator, PPAR-γ orchestrates fatty acid storage, transport, and metabolism, as well as glucose metabolism, by modulating gene expression in adipocytes, hepatocytes, and other types of cells.	[28,31]
The caspase pathway	After cleavage by caspase-1 (CASP1), PPAR-γ translocates to mitochondria, where it binds to and inhibits medium-chain acyl-CoA dehydrogenase (MCAD).	In this pathway, PPAR-γ acts as an endogenous protein inhibitor. Its inhibitory activity promotes lipid droplet accumulation by reducing the oxidation rate of fatty acids via suppression of medium-chain acyl-CoA dehydrogenase (MCAD).	[32]
The nuclear factor κB (NF-κB) signaling pathway	PPAR-γ ubiquitinates the p65 subunit of NF-κB, targeting it for proteasomal degradation. Additionally, PPARγ upregulates the gene encoding the inhibitory subunit IκBα, further suppressing NF-κB activity.	In this pathway, PPAR-γ exhibits a dual regulatory role in modulating NF-κB signaling. Firstly, it acts as an E3 ubiquitin ligase, directly ubiquitinating the p65 subunit of NF-κB and targeting it for proteasomal degradation. Secondly, PPAR-γ functions as a transcription factor, inducing IκBα gene expression to enhance NF-κB inhibition. By suppressing the NF-κB pathway through these mechanisms, PPAR-γ attenuates inflammatory responses in affected cells.	[33,34]
The activator protein 1 (AP1) signaling pathway	PPAR-γ binds to Jun-Fos heterodimers of the transcription factor AP1, preventing their interaction with DNA. Additionally, PPAR-γ sequesters the transcriptional coactivator of AP1, CREB-binding protein (CBP), further suppressing AP1-dependent gene expression.	In this pathway, PPAR-γ exhibits a dual regulatory role in modulating AP1 activity. Firstly, it acts as a transcriptional repressor of Jun-Fos heterodimers of the transcription factor AP1, blocking their ability to bind DNA and activate target genes. Secondly, PPAR-γ competes with AP1 for shared transcriptional coactivators (e.g., CBP/p300), disrupting the capacity of AP1 to form functional transcriptional complexes. The impact of transrepression depends on the context. For instance, it reduces the expression of pro-inflammatory cytokines (e.g., TNF-α, IL6, and IFN-γ) in immune cells.	[35,36,37]
The signal transducer and activator of transcription (STAT) signaling pathway	The heteromers of PPAR-γ STAT1 induce the gene of the scavenger receptor complex of differentiation 36 (CD36). In turn, the heteromers of PPAR-γ and STAT6 upregulate the expression of the fatty acid transporter, fatty acid-binding protein 4 (FABP4).	In this pathway, PPAR-γ acts as a transcriptional coactivator of STAT1 and STAT6, driving changes in gene expression that promote lipid accumulation. By inducing CD36 and FABP4, the heteromers of PPAR-γ with STAT proteins enhance lipid uptake and storage. These alterations result in the accumulation of lipid droplets within affected cells.	[38,39,40]
The MAPK pathway	Mitogen-activated protein kinases (MAPKs), including extracellular signal-regulated kinases (ERKs) and c-Jun N-terminal kinase (JNK), directly phosphorylate PPAR-γ, modulating its transcriptional activity.	In this pathway, PPAR-γ serves as an effector protein of the named MAPKs. Phosphorylation by these kinases reduces the transcriptional activity of PPAR-γ, leading to suppression of its target genes (e.g., the genes controlling lipid metabolism).	[41,42]

**Table 2 cells-14-00534-t002:** Clinical trials of PPAR-γ agonists in melanoma and advanced cancers with enrollment of melanoma patients.

Therapeutic Agent	Trial ID	Phase	Status	*N*	Brief Description	Objectives and Outcome	Ref.
Anti-PD-L1 immunotherapy (PD-L1—programmed death ligand 1) alone or in combination with either metformin or rosiglitazone (RSG)	NCT04114136	II	A	72	Controlled, randomized, open clinical trial, with parallel groups, without using placebo, three treatment arms	This study aimed to examine whether metformin and RSG will reduce oxygen consumption by tumors, creating a less hypoxic environment for T cells, with pharmacologic remodeling of the tumor microenvironment (TME) to restore the effector function of T cells, acting synergistically with anti-PD-L1 monoclonal antibodies to produce a higher response rate than conventional anti-PD-L1 immunotherapy.	not published
Bexarotene in combination with RSG		I	C	45	Open-label pilot study	This clinical trial investigated potential synergistic effects of combining bexarotene and RSG in patients with refractory solid tumors. Although the combination demonstrated safety and feasibility in heavily pretreated patients, it failed to induce objective tumor responses.	[55]
Temsirolimus in combination with pioglitazone (PGZ), etoricoxib, and metronomic low-dose trofosfamide versus DTIC (dacarbazine)	EUCTR2011- 002611-29-DE/NCT01614301	II	C	48	Controlled, randomized, open clinical trial, with parallel groups and two treatment arms, without using placebo	This study aimed to evaluate overall survival, objective response rate, time to progression, time to partial response, quality of life, tolerability, and safety of a proposed therapeutic regimen in patients with chemorefractory malignancies. The study found that the proposed therapeutic regimen was active and well tolerated by patients with chemorefractory malignancies.	[56]
Bexarotene in combination with efatutazone (EFZ)	NCT01504490	I	T	9	Controlled, randomized, open clinical trial, with parallel groups and unspecified number of treatment arms without using placebo	This study aimed to evaluate the safety and efficacy of a novel drug combination, EFZ and bexarotene, in patients with advanced cancers. The trial was designed to explore the potential synergistic effects of these agents in modulating the tumor microenvironment and improving therapeutic outcomes. However, the study was prematurely terminated following the withdrawal of EFZ from the market, preventing further evaluation of its clinical potential.	not published
EFZ	NCT00408434	I	C	32	Open-label dose escalation study	This phase I study aimed to evaluate the safety, tolerability, and preliminary efficacy of EFZ, a novel oral PPAR-γ agonist, in patients with advanced or metastatic solid tumors. The results demonstrated that EFZ has an acceptable tolerability with evidence of disease control in patients with advanced malignancies.	[57]
EFZ	NCT00881569	I	T	2	Long-term follow-up open-label safety extension study in non-randomized, observational phase	This study was designed to allow participants who completed a prior clinical trial of EFZ without disease progression or unacceptable toxicity to continue receiving the study drug. The study was terminated after both enrolled patients experienced disease progression, indicating loss of therapeutic benefit.	not published
ω3 PUFAs	NCT01032343	N/A	C	79	Double-blind randomized, placebo-controlled nutritional study with two treatment arms	This study investigated whether dietary ω3 polyunsaturated fatty acids (PUFAs), particularly eicosapentaenoic acid (EPA), could mitigate UV-induced cutaneous immunosuppression in healthy human subjects without prior skin cancer diagnoses. The results showed that the dietary supplement used in this study shifted eicosanoid synthesis towards less pro-inflammatory bioactive components, promoting a regulatory milieu under basal conditions and in response to the inflammatory insult.	[58]
Resveratol	NCT00098969	I	A	40	Open-label dose escalation study	This clinical trial aimed to assess the safety, tolerability, and optimal dosing of resveratrol in healthy individuals, with secondary exploration of its chemopreventive potential. The results demonstrated that resveratol exhibits a potential chemopreventive effect, by reducing the plasma levels of circulating insulin-like growth factor 1 (IGF1) and insulin growth factor-binding protein 3 (IGFBP3).	[59]
Curcumin	NCT01201694	I	C	28	Open-label, non-randomized, dose-escalation study with single group assignment and unspecified number of treatment arms without using placebo	This clinical trial aimed to evaluate the safety, tolerability, and maximum tolerated dose of surface-controlled water-dispersible curcumin in patients with advanced solid tumors.	not published
Curcumin	NCT02138955	IB	C	32	Open-label, non-randomized, dose-escalation study with single group assignment and seven treatment arms without using placebo	This clinical trial aimed to evaluate the safety, tolerability, and maximum tolerated dose of liposomal curcumin in patients with advanced or metastatic solid tumors. The results showed that liposomal curcumin was safe and well tolerated by patients with no dose-limiting toxicities observed, supporting its potential for therapeutic use.	[60]
Curcumin in combination with cilazapril or losartan, aliskerin, propranolol, aspirin, and metformin	ACTRN12619001078145	II	C	50	Open-label, non-randomized, non-blinded, single treatment arm without using placebo	This clinical trial aimed to evaluate the tolerability and preliminary efficacy of a novel therapeutic regimen targeting the renin–angiotensin system and its downstream signaling pathways in patients with advanced solid tumors.	not published

*N*—number of participants; A—active; C—completed; T—terminated; N/A—not applicable.

**Table 3 cells-14-00534-t003:** Selected ligands of PPAR-γ: properties and limitations.

Ligand	Specificity	Mode of Action	PPAR-y Independent Effects	Limitations
CGZ	CGZ is a potent and specific PPAR-γ agonist, with a selectivity of over 33-fold compared to other PPARs [128].	CGZ primarily exerts its effects through the activation of PPAR-γ, resulting in enhanced adipogenesis and fat storage, improved insulin sensitivity, and altered hormone production in adipose tissue, as well as potential anticancer properties.	CGZ downregulates anti-apoptotic proteins such as Bcl-2 and Bcl-xL (Bcl—B-cell lymphoma) increasing the apoptotic rate of cultured cancer cells [129]. CGZ promotes the ubiquitin-dependent proteasomal degradation of several key protein regulators involved in cell cycle progression and cancer cell survival, such as cyclin D1 (CCND1) and flice-inhibitory protein (FLIP) [130]. CGZ induces cell cycle arrest in various cancer cell lines, including melanoma cells [64].	Potential severe side effects (based on what we know about other TZDs) [131,132] Limited use—used in research for its anti-inflammatory and anti-fibrotic properties [133], and its role as a PPAR-γ agonist. Limited data on drug interactions. Never marketed. Lack of clinical use.
PGZ	PGZ binds with high affinity to PPAR-γ, a property crucial for its insulin-sensitizing effects [134]. Additionally, it exhibits weak agonistic activity toward PPAR-α [135].	PGZ primarily acts as a selective agonist of PPAR-γ, with secondary effects on PPAR-α. It improves insulin sensitivity, reduces hepatic glucose production, and enhances adipokine secretion. PGZ effectively lowers blood glucose levels and improves lipid profiles in patients with T2D.	Due to its interactions with astrocytes and glial cells in the spinal cord, PGZ rapidly reduces neuropathic pain within minutes [136]. PGZ exhibits anti-inflammatory effects, such as suppressing vascular cell adhesion molecule 1 (VCAM1) in endothelial cells, through the activation of PPAR-α [137]. PGZ inhibits the growth of cancer cells by suppressing the STAT3 signaling pathway and enhancing the expression of the apoptosis-inducing factor (AIF) [138]. PGZ attenuates platelet-derived growth factor (PDGF)-induced proliferation of vascular smooth muscle cells through both AMPK-dependent and AMPK-independent mechanisms (AMPK—AMP-dependent protein kinase). Furthermore, it suppresses the mTOR/p70S6K and ERK signaling pathways in these cells, a process essential for their proliferation (mTOR—mechanistic target of rapamycin; p70S6K—p70 ribosomal S6 kinase) [139]. PGZ reduces oxidative stress by modulating the phosphorylation of SHC and inhibiting the activity of PKC-β (SHC—Src homology and collagen; PKC-β—protein kinase C-β) [140].	Contraindications for patients with NYHA class III or IV heart failure (NYHA—New York Heart Association). Serious side effects of PGZ include dose-related fluid retention and edema [29934374], as well as an increased risk of bladder cancer, particularly in long-term users [141,142]. Increases the risk of bone fractures, especially in postmenopausal women [143]. Ocular effects, such as macular edema, potentially causing blurred vision or vision loss [144].
RSZ	RRSZ exhibits high specificity for PPAR-γ compared to other PPAR subtypes, showing no significant activity at PPAR-α or PPAR-β/δ [145,146].	RSZ acts as a potent PPAR-γ agonist, enhancing insulin sensitivity and glucose metabolism by modulating the transcription of insulin-responsive genes involved in glucose production, transport, and utilization [145,146].	RSZ reduces oxidative stress by enhancing the expression of antioxidant enzymes, such as glutathione peroxidase (GPX) and superoxide dismutase (SOD) [147]. RSZ alters mitochondrial function, affecting their Red/Ox potential. [148]. RSZ inhibits glucose-induced oxidative stress by reducing NAD(P)H oxidase activity through the activation of AMPK [149]. RSZ activates the glucocorticoid receptor [150]. RSZ suppresses the fibrotic response in fibroblasts by modulating the p38 MAPK pathway [151].	Contraindications for using RSZ in patients with NYHA class III and IV heart failure, active liver disease, or diabetic ketoacidosis. Serious side effects of RSZ include increased cardiovascular risks for patients with heart failure—when combined with insulin, 27% higher risk of stroke, compared to PGZ [152,153]; weight gain and edema [154]; and hepatotoxicity—in patients with pre-existing liver conditions [155]. Regulatory actions: The FDA restricted RSZ in 2010 due to cardiovascular concerns. Sales resumed normally in 2013 after restrictions were lifted.
TGZ	Compared to RSZ and PGZ, TGZ exhibits a lower affinity for PPAR-γ. This difference is reflected in the higher clinical dosages required for TGZ (400–800 mg/day) compared to RSZ (4–8 mg/day) and PGZ (15–45 mg/day). Moreover, TGZ displays weaker binding activity to PPAR-α [156].	TGZ primarily acts as a selective PPAR-γ agonist, inducing genes involved in glucose and lipid metabolism, thereby enhancing insulin sensitivity in muscle and adipose tissue while reducing hepatic gluconeogenesis [156]. However, the metabolic biotransformation of TGZ into reactive metabolites causes idiosyncratic hepatotoxicity, which led to withdrawal of TGZ from the market [156,157].	TGZ induces apoptosis in cancer cells by disrupting the interactions between anti-apoptotic and pro-apoptotic proteins [158,159]. TGZ causes cellular acidosis by inhibiting the Na^+^/H^+^ exchanger, resulting in reduced acid extrusion from the cell. [160]. TGZ reduces alanine aminotransferase (ALT) activity [161]. TGZ reversibly inhibits the contraction of vascular smooth muscle cells [162]. TGZ decreases the expression of inflammatory mediators [159].	In 2000, TGZ was withdrawn from the market due to concerns about severe hepatotoxicity and other serious side effects (peripheral edema and hypoglycemia) [69]. TGZ not studied in patients with NYHA class III/IV heart failure) Increased risk of hypoglycemia [163,164].
EFZ	EFZ demonstrates a significantly higher affinity for PPAR-γ compared to other PPAR subtypes, establishing it as a highly selective agonist. Furthermore, EFZ is considerably more potent than RSZ and TGZ in activating PPAR-γ-mediated transcription.	EFZ selectively activates PPAR-γ, distinguishing itself from other TZDs by downregulating the phosphorylation of PKB (a key enzyme in the prosurvival pathway) without affecting the phosphorylation of ERK. [57,100,165].	EFZ reduces the expression of COX2 and IL8, which are key drivers of inflammation and angiogenesis in tumors, thereby potentially mitigating these processes that contribute to tumorigenesis. EFZ exerts an additive effect on the inhibition of NF-κB signaling, leading to a reduction in the expression of pro-inflammatory cytokines, such as IL6 and TNF. EFZ enhances the activity of the nuclear receptor LXR-α (liver X receptor α), which plays a crucial role in the metabolism of cholesterol. This synergistic effect promotes the efflux of cholesterol from cancer cells, inducing their apoptosis. EFZ induces ketogenesis by upregulating the ketogenic enzyme HMGCS2. Additionally, EFZ suppresses pyruvate dehydrogenase kinase 4 (PDK4) and inhibits β-oxidation, thereby shifting the energy metabolism away from the Warburg effect. This metabolic shift leads to the generation of reactive oxygen species (ROS) and results in cell cycle arrest [57,98].	In 2012, EFZ was discontinued by the manufacturer due to slow enrollment and limited efficacy as a single agent in clinical trials. Adverse effects (fluid retention—edema and pleural effusion), hematologic effects (neutropenia, leukopenia, and anemia) [57,100,166]. Exclusion of diabetic patients due to increased fluid retention risks [100]. Dose-limiting toxicities [57]. Limited efficacy in advanced cancers: partial response [57]. Stabilization of the disease with lack of reasonable improvement [57,100]. Lack of clinical validation beyond phase II.
MM902	MM902 specifically and irreversibly interacts with PPAR-γ [117].	MM902 functions as a selective and irreversible antagonist of PPAR-γ, inhibiting its activity and thereby affecting cellular processes related to tumorigenesis and metabolism [117].	MM902 demonstrates weak interactions with a mutant form of BRAF (V600E) and IKK-α, both potentially beneficial for melanoma patients [117]	Irreversible binding to PPAR-γ. Lack of clinical translation.
T0070907	T0070907 exhibits an 800-fold selectivity to PPAR-γ over PPAR-α and PPAR-β/δ.	T0070907 operates as a selective and irreversible antagonist of PPARγ, with additional effects that extend beyond PPARγ inhibition. Moreover, in vitro studies found that T0070907 decreased the phosphorylation of PPAR-γ [123].	T0070907 inhibits the phosphorylation of ERK and focal adhesion kinase (FAK) [123]. T0070907 inhibits the proliferation of breast cancer cells with a dominant negative form of PPAR-γ. In the same cells, T0070907 significantly reduces their migration capabilities in wound-healing assays and transwell migration assays [123]. In cervical cancer cells, T0070907 induces G2/M cell cycle arrest and mitotic catastrophe by reducing the levels of α- and β-tubulin [122].	PPAR-y-independent cytotoxicity [167]. Irreversible covalent binding [120]. Lack of clinical translation.
GW9662	GW9662 is 100–1000 times more specific to PPAR-γ than PPAR-α and PPAR-β/δ [168,169].	GW9662 acts as a covalent antagonist of PPAR-γ, irreversibly modifying its ligand-binding domain [120].	GW9662 inhibits the growth and survival of breast cancer cells independently of its action on PPAR-γ [169]. In macrophages, GW9662 enhances lipogenesis, and the accumulation of triglycerides by activating PPAR-β/δ-mediated signaling [170]. In human T-helper cells, GW9662 does not interfere with the immunomodulatory effects of PPAR-γ activated by fatty acids. Instead, it produces an additive inhibitory effect on the biosynthesis of IL2 [171].	PPAR-y-independent cytotoxicity (≥10 μM). Cross-reactivity with PPAR-α and -β/δ [125]. Irreversible covalent binding. Lack of transcriptional repression [120]. Lack of clinical translation.

## Data Availability

This review article does not mention, cite or analyze unpublished original data. All cited papers are available from the publishers’ websites or public domains. Data sharing is not applicable to this study.

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
