# Peer review of "PPAR-γ in Melanoma and Immune Cells: Insights into Disease Pathogenesis and Therapeutic Implications"

_cells, 2025, doi:10.3390/cells14070534_

Round 1

Reviewer 1 Report

Comments and Suggestions for Authors

The authors report what is currently known about the role of PPARg in melanoma in terms of its biological effects and therapeutic implications. The review is divided into three parts: The first part describes the initiation and progression of melanoma; the second describes the different PPARg ligands and their applications in cancer field, as well as and the co-activators and co-repressors essential for the regulation of transcriptional activity; the third and final part which discusses more specifically the biological effects linked to the activation of PPARg in melanoma cells and the modulation of the TME, highlighting the possibility of exploiting these effects to develop new therapeutic strategies. The authors provide a comprehensive review of the literature reporting in vitro and clinical trial data also on the use of PPARg ligands in tumours other than melanoma, highlighting the complexity of the underlying molecular mechanisms, which are often independent of PPARg activation and which, although showing good potential as anti-tumour agents, require further information.

However, I suggest to revise some aspects:

  • For greater clarity and ease of reading of Figure 4, I recommend dividing the figure into three boxes, one with the TDZs agonists, one with the non-TDZs and another with the endogenous agonists. I also recommend writing the names of compounds under their chemical formula rather than letters.
  • Line 209: add troglitazone before TGZ. I think this is the first time it has been mentioned.
  • In Figure 5, as in the case of the Figure 4, I recommend writing the name of the different compounds below the chemical structure.
  • Add in the text the Figure 5
  • Figure 6: delete (a) MM902; (b) T0070907; (c) GW9662. They are clearly indicated under the chemical structures and (a), (b) and (c) don't appear in the figure.
  • Lines 426-436 report general considerations about PPARg agonists in cancer treatment. In my opinion, they would be more appropriate in the Conclusion section and therefore need to be moved. The MM902 paragraph can start on line 437.
  • Figure 7 needs some modifications. It is necessary to add a legend indicating the significance of blue circle, green circle, and red square. Substitute or add to “NCoR, SMRT” the word “co-repressors” and similarly substitute or add to “PGC1a” the word “co-activators” for a simpler and more immediate reading of the figure. SRC needs to be added to the (c) description of the figure or deleted from the (c) cartoon. Replace (h) in the figure description with (i) indicated in the cartoon. Consider also to revised the cartoon in order to clarify and make more clearer the different steps. For example, add “agonist-receptor complex” in the corresponding cartoon as a very small description of the specific step.
  • Lines 708-714 This section discusses the effects of combined calcipotriol and phototherapy in vitiligo. I don't think it is relevant to the topic of the paragraph (PPARg and pigmentation in melanoma) and I think these lines need to be deleted. Otherwise, if the authors think they are relevant, they probably need to better explain the concept they wanted to express.
  • Line 822 add tumor microenvironment before TME
  • Revise the Conclusion section according to the point 6. In line 1015-1019, the authors talk about the need to carry out further studies on PPARg ligands and their effects in order to be able to use them in clinical practice for different dermatological pathologies. In my opinion, it is more appropriate to make the same type of considerations with regard to melanoma, which is the subject of the review. Therefore, I recommend revising this last part.

Author Response

Dear Editor,

We would like to confirm that we have received three opinions regarding the submitted manuscript "PPAR-γ in Melanoma and Immune Cells: Insights into Disease Pathogenesis and Therapeutic Implications" (Ref. # cells-3542996). To comply with MDPI rules, we will respond to each reviewer separately, addressing their concerns in the order they were raised.

Reviewer #1 made ten comments:

We would like to thank Reviewer #1 for the detailed analysis of the text and for the suggestions that the reviewer made. The reviewer’s comments have allowed us to significantly improve the manuscript, especially the figures, as well as the descriptive part of the main text.

COMMENT #1: For greater clarity and ease of reading of Figure 4, the reviewer recommended dividing the figure into three boxes: one with the TZD agonists, one with the non-TZD agonists, and another with the endogenous agonists. The reviewer also recommended writing the names of the compounds under their chemical formulas rather than using letters.

RESPONSE #1: In response to Reviewer's comment, we have revised Figure 4 by dividing it into three separate sections: one for thiazolidinediones, one for non-thiazolidinedione agonists, and one for endogenous PPAR-γ agonists. As suggested, we have boxed each part of the figure to improve clarity and readability. Additionally, we have displayed the names of the PPAR-γ agonists under their respective chemical formulas within the figure, and consequently, removed them from the figure caption to avoid redundancy.

COMMENT #2: The reviewer suggested that we add the word "troglitazone" before "TGZ" (line 209), since, in accordance with the publisher's rules, this was the first time the term was mentioned in the text.

RESPONSE #2: We have taken Reviewer's suggestion into consideration and revised the text accordingly. Specifically, we have added the word "troglitazone" before "TGZ" in the revised manuscript (line 227), which now reads: "However, it played a significant role in the development and marketing of more potent insulin sensitizers, such as pioglitazone (PGZ), rosiglitazone (RGZ), and troglitazone (TGZ)".

COMMENT #3: Similar to Figure 4, the reviewer suggested that we add the names of the compounds mentioned in the figure caption below their respective chemical structures.

RESPONSE #3: In response to Reviewer #1's comment regarding Figure 5, we have revised the figure by transferring the names of the compounds from the figure caption to the figure itself, where they are now displayed under their respective chemical formulas, consistent with the format used in Figure 4.

COMMENT #4: Reviewer #1 suggested that we mention Figure 5 in the main text of the manuscript.

RESPONSE #4: In response to Reviewer's suggestion, we have added a reference to Figure 5 in the main text of the manuscript. Specifically, we have revised the text to include a mention of Figure 5 in line 402 of the revised manuscript, which now reads: "Moreover, many bioactive compounds derived from plants may potentiate the transcriptional activity of PPAR-γ acting as its natural ligands [103] (Figure 5)".

COMMENT #5: Reviewer #1 also suggested that we revise the caption of Figure 6. Specifically, the reviewer recommended that we delete the names of the compounds from the caption, as they are clearly indicated under the chemical structures. Additionally, the reviewer suggested removing the letters "a", "b", and "c" (in parentheses) from the caption, as they do not appear in the figure.

RESPONSE #5: In response to Reviewer's comment, we have revised the caption of Figure 6 as suggested. Specifically, we have removed the letters "a", "b", and "c" (in parentheses) and the names of the compounds from the caption, as they are already clearly indicated under the chemical structures in the figure.

COMMENT #6: Reviewer #1 recommended that we move lines 426-436 from the former subsection "2.1.9.1 MM902" to the "Conclusion", as they contain general considerations about PPARγ agonists in cancer treatment. The reviewer suggested starting the subsection dedicated to MM902 with the next paragraph.

RESPONSE #6: In response to Reviewer's comment, we have relocated the text fragment (formerly lines 426-436) to the "Conclusion" section (now lines 1,102-1,115 in the revised version, which begins on page 32). We have also made the necessary adjustments to the preceding and following paragraphs to ensure a smooth integration of the relocated text.

COMMENT #7: Reviewer #1 suggested that we modify the former Figure 7, which is now Figure 8 in the revised version of the manuscript. The reviewer's suggestions can be summarized as follows:

  1. The reviewer requested that we explain the significance of the blue circle, green circle, and red square in the figure caption.
  2. The reviewer suggested that we add the word "co-repressors" after "NCoR, SMRT" or replace the words "NCoR, SMRT" with the word "co-repressors".
  3. The reviewer recommended that we add the word "co-activators" after "PGC1-α" or replace the word "PGC1-α" with the word "co-activators" to facilitate a simpler and more immediate reading of the figure.
  4. The reviewer requested that we either add "SRC" to part (c) of the figure caption or delete it from the (c) cartoon.
  5. The reviewer suggested that we replace "(h)" with "(i)" in the figure caption to match the labeling in the cartoon.
  6. The reviewer recommended that we revise the cartoon to clarify the different steps, specifically by indicating the presence of the "agonist-receptor complex" as a descriptive label for a particular step.

RESPONSE #7: In response to Reviewer's comment, we have prepared a revised version of Figure 8 (formerly Figure 7) and modified its caption to address the reviewer's concerns.

Regarding the first point of Comment #7, we have explained the meanings of the blue circle, red square, and green circle in the revised figure caption. Specifically, we have clarified that the blue circle represents a specific agonist of RXR-α that interacts with the molecule of RXR-α (line 606), the green circle represents an agonist that interacts with PPAR-γ (lines 610-611), and the red square represents an antagonist that interacts with the molecule of PPAR-γ (line 617).

In response to the second point of Comment #7, we have complied with the reviewer's request by adding the term "Co-repressors:" before "NCoR, SMRT" in the cartoon, and have placed this term on a preceding line to clarify its meaning.

Regarding the third point of Comment #7, we have also complied with the reviewer's request by adding the term "Co-activators:" before the names of proteins that increase the transcriptional activity of PPAR-γ (PGC1a, SRC, p300, and CBP) as indicated in part (d) of the cartoon.

In response to the fourth point of Comment #7, we renamed "SRC" to “NCoA1” in the figure and part (c) of the caption (line 614), specifically stating "(c) Then, PPAR-γ recruits NCoA1" to avoid confusion between steroid receptor coactivator and sarcoma family member, which are both abbreviated as “SRC".

Regarding the fifth point of Comment #7, we have replaced "(i)" with "(h)" in the cartoon to maintain consistency with the standard English alphabet.

Finally, in response to the sixth point of Comment #7, we have designated the "agonist-receptor complex" as "AgRC" and the "antagonist-receptor complex" as "AnRC" in parts (a) and (e) of the cartoon, and have adjusted the figure caption accordingly. The AgRC is mentioned in line 611, where we state "...to form an agonist-receptor complex (AgRC)", and the AnRC is mentioned in lines 617-618, where we state "leads to the formation of antagonist-receptor complex (AnRC) and stabilizes PPAR-γ in the 'closed' conformation".

COMMENT #8: Reviewer #1 recommended that we delete lines 708-714 from the section "3.2. The role of PPAR-γ in pigmentation" as they appear to be irrelevant, or alternatively, provide an explanation for their necessity.

RESPONSE #8: In response to Reviewer's comment, we have revised the section "3.2. The role of PPAR-γ in pigmentation" to address the concern. We have clarified the relevant text and incorporated the necessary information into the last paragraph of this section, which can be found in lines 807-814 (page 27) of the revised manuscript.

COMMENT #9: Reviewer #1 suggested that we add the words "tumor microenvironment" before the abbreviation "TME", as it was the first instance of its use in the manuscript (line 822).

RESPONSE #9: In response to reviewer’s comment we would like to acknowledge that revising the manuscript in accordance with reviewers’ suggestions we made significant changes in its body. For this reason, the term “tumor microenvironment” now appears earlier (page 12, lines 317-318) in the phrase “According to Konger et al. [82], treatment with RGZ (8 mg/ kg/ day) reduces the number of myeloid-derived suppressor cells (MDSCs) associated with tumor microenvironment (TME).” Respectively, we put the abbreviation “TME” in that place instead of one suggested by the reviewer.

COMMENT #10: Reviewer #1 recommended that we rewrite the Conclusion section, presumably in accordance with the concerns raised in COMMENT #6. The reviewer's suggestions can be summarized as follows:

  1. The reviewer recommended rewriting the Conclusion section in accordance with point 6, which is also related to the concern expressed in COMMENT #6.
  2. In lines 1015-1019, where we discussed the need for further studies on PPAR-γ, the reviewer suggested that we make similar considerations regarding melanoma, which is the primary subject of the review.

RESPONSE #10: In response to Reviewer's comment, we have made revisions to the Conclusion section to address the concerns raised by the reviewer.

Regarding PART A of the comment, we have integrated the text fragment (formerly lines 426-436) into the Conclusion section, as previously discussed in response to COMMENT #6. We have also made adjustments to the preceding and following paragraphs in this section to ensure a smooth and logical flow of ideas.

Regarding PART B of the comment, we have revised the text in line 1,134 (page 33) to better align with the focus of the review on melanoma. The revised sentence now reads: "However, our current understanding and experience with PPAR-γ and its ligands remain insufficient to confidently recommend these agents for the treatment of melanoma." This revision aims to provide a more nuanced and relevant conclusion regarding the potential application of PPAR-γ and its ligands in the context of melanoma.

We're grateful for the opportunity to revise and resubmit.

Sincerely yours,
MEZENTSEV, Alexandre, PhD

Reviewer 2 Report

Comments and Suggestions for Authors

The review paper submitted for review entitled: "PPAR-γ in Melanoma and Immune Cells: Insights into Disease 2

Pathogenesis and Therapeutic Implications' is an extensive source of literature data that touches on an important topic, i.e. melanoma, which due to its etiology is an important topic that should be constantly learned... because although we know a lot, it is still too little.

Although the paper contains all the elements that are required for a review paper, the amount of information contained in it is quite difficult to quickly grasp.

Firstly, the subsections are very poorly / not very visible, which very often causes the thread of the topic to be lost

Secondly, the paper lacks figures / tables that would collect the most important information regarding a given element, e.g. subsection - The ligands of PPAR-γ, their specificity and limitations, it begs for a summary table to make the paper legible and valuable

Please review and eliminate bold, e.g.: caption of Figure 4

Author Response

Dear Editor,

We would like to acknowledge receipt of the three reviews for our submitted manuscript, "PPAR-γ in Melanoma and Immune Cells: Insights into Disease Pathogenesis and Therapeutic Implications" (Ref. # cells-3542996). In accordance with MDPI guidelines, we will respond to each reviewer's comments individually, addressing them in the order they were raised.

Before proceeding with our responses, we would like to express our sincere gratitude to Reviewer #2 for the insightful suggestion to restructure the section of the manuscript dedicated to the expression and regulation of PPAR-γ. We have made a concerted effort to address this concern and appreciate the opportunity to revise our work accordingly. We are also deeply thankful for Reviewer's meticulous and attentive reading of our paper, as well as for the thoughtful comments, which we believe have significantly enhanced the revised version of the manuscript.

In our assessment the reviewer made 3 comments.

COMMENT #1: The reviewer noted that the subsections are not very visible, which often causes the thread of the topic to be lost. The reviewer suggested summarizing the key findings in tables to improve clarity.

RESPONSE #1: In response to Reviewer's comment, we have restructured the section dedicated to the expression and regulation of PPAR-γ to improve clarity and visibility. We provide an introductory overview of this section in the paragraph spanning lines 174-185, which precedes the subsection "2.1. The ligand-dependent activation of PPAR-γ" (located at the bottom of page 7 in the revised manuscript).

To further enhance the readability and organization of the manuscript, we have added three tables that summarize key findings. Table 1 provides a concise overview of the intracellular signaling pathways in which PPAR-γ is involved, and can be found on pages 5-6, following the first paragraph of subsection 2.1. Table 2 summarizes the available information on clinical trials studying PPAR-γ agonists, and is located on pages 8-10, immediately after the subsection "2.1.1 Thiazolidinediones". Table 3 reviews the data on specificity, mode of action, limitations, and PPAR-γ-independent agonists and antagonists, and is presented on pages 16-20.

We have also made concluding remarks in the subsection "2.3 PPAR-γ ligands as a therapeutic strategy" (page 24), where we introduce Table 3, to provide a clear summary of the key points and takeaways from this section. These revisions aim to address the reviewer's concern and improve the overall clarity and flow of the manuscript.

COMMENT #2: Reviewer #2 suggested including figures or tables that collect the most important information regarding a given element, such as the subsection "The ligands of PPAR-γ, their specificity and limitations".

RESPONSE #2: As we mentioned earlier in response to the previous comment, we have introduced Table 3, which compiles the data on specificity, mode of action, limitations, and PPAR-γ-independent properties exhibited by agonists and antagonists (pages 16-20). Additionally, we have provided brief concluding remarks in the subsection "2.3 PPAR-γ ligands as a therapeutic strategy" (page 16), which summarize the key points related to the ligands of PPAR-γ, their specificity, and limitations, thereby addressing Reviewers suggestion to include a concise collection of important information on this topic.

COMMENT #3: Reviewer #2 suggested that we refrain from using bold font in the captions of figures, such as Figure 4.

RESPONSE #3: In response to this request, we have carefully reviewed all figure captions and ensured that they no longer utilize bold formatting, thereby conforming to the reviewer's preference.

Thank you for considering our revised manuscript.

Sincerely yours,
MEZENTSEV, Alexandre, PhD

Reviewer 3 Report

Comments and Suggestions for Authors

The authors review the role of PPARG in melanoma and immune system.

This review is very comprehensive, but it could be improved tremendously by adding additional figures and tables:

Please add a figure with a schematic and a table with all the possible pathways PPARG is involved with including relevant references.

Also add a table with all the agonists and antagonists, their specificity, limitations, method of action and their usage in the clinic/clinical trials (including the ClinicalTrials.gov ID) and relevant references.

Additionally, the authors should show a schematic with PPARG domains depicting ligand-independent activation sites, and sites of interaction with Figure 8 agents if known

Figure 7: please add more cellular components depicting the cytosol, the nuclear translocation and DNA binding in a more visual way, what are the red square and green and blue circles?

The schematic has no “h” but an “i”. The “closed” and “open” conformations are not evident from the schematic.

Figure 8: Please add PPARG to the figure.

Author Response

Dear Editor,

We would like to acknowledge receipt of the three reviews for our submitted manuscript, "PPAR-γ in Melanoma and Immune Cells: Insights into Disease Pathogenesis and Therapeutic Implications" (Ref. # cells-3542996). Before responding to the comments and questions posted by Reviewer #3, we would like to express our sincere gratitude to the reviewer for the time and effort devoted to evaluating our manuscript. We also appreciate the reviewer's thoughtful questions regarding the signaling pathways involving direct participation of PPAR-γ, and we have made a concerted effort to address these concerns in our revised manuscript.

COMMENT #1: Reviewer #3 recommended that we add a figure with a schematic and a table that includes all the possible pathways in which PPAR-γ is involved, along with relevant references.

In response to Reviewer's comment, we carefully considered the request to add a figure with a schematic and a table outlining the possible pathways involving PPAR-γ. After thorough examination, we concluded that creating a single figure to encompass all intracellular signaling mechanisms would be technically feasible but potentially overwhelming due to the complexity and multitude of involved details. We were concerned that such a figure might be difficult to navigate and interpret.

Instead, we believed that summarizing the available information about these signaling pathways in a table format would be a more effective and reader-friendly approach. This would allow readers to easily grasp the complexity of the crosstalk between these pathways and form their own judgments. Therefore, we introduced Table 1, which provides a concise summary of the signaling pathways in which PPAR-γ participates (located on pages 5-6, following the first paragraph of subsection 2.1). We hope that this table will provide a clear and accessible overview of the relevant information, and we appreciate Reviewer's suggestion, which prompted us to reconsider our approach to presenting this complex information.

COMMENT #2: Reviewer #3 suggested that we add a table that includes all the agonists and antagonists of PPAR-γ, their specificity, limitations, mode of action, and their usage in clinical trials (including the ClinicalTrials.gov ID), along with relevant references.

In response to Reviewer's comment, we have prepared two separate tables to address the reviewer's concerns. Table 2 focuses on the clinical aspects summarizing the available information on clinical trials that have examined the safety and therapeutic effects of PPAR-γ agonists (please see pages 8-10, following the subsection "2.1.1 Thiazolidinediones"). Table 3, on the other hand, addresses the experimental aspects, discussing the data on specificity, mode of action, limitations, and PPAR-γ-independent effects of PPAR-γ agonists and antagonists (pages 16-20).

By splitting this information into two tables, we aimed to highlight the importance of both clinical and experimental studies, and to facilitate a clearer understanding of the complex relationships between these factors. We believe that this approach will enable readers to more easily follow and appreciate the significance of the information presented in Tables 2 and 3, and we appreciate Reviewer's suggestion, which prompted us to reorganize and clarify this critical information.

COMMENT #3: The reviewer wanted us to include a scheme illustrating the PPAR-γ domains and indicate their ligand-dependent and ligand-independent activation sites, as well as the sites of interaction with Figure 8 agents if known.

In response to this comment, we have prepared a new Figure 7, which provides a schematic illustration of the PPAR-γ domains. The figure caption highlights the key activatory functions of PPAR-γ (AF-1 within the A/B domain and AF-2 within the E/F domain), and clarifies their roles in both ligand-independent and ligand-dependent modulations of PPAR-γ transcriptional activities, respectively. To further elucidate the ligand-independent mechanisms, we have identified the sites of various post-translational modifications, such as acetylation, phosphorylation, SUMOylation, and β-O-linked N-acetylglucosamine (O-GlcNAc) modification, within the PPAR-γ molecule. Additionally, we have expanded the subsection "2.4 Ligand-independent changes in the transcriptional activity of PPAR-γ" (lines 540-596, pages 20-22) to provide more comprehensive information on the post-translational modifications of PPAR-γ and their regulatory effects, as suggested by the reviewer.

COMMENT #4: In Figure 7 (now revised as Figure 8, page 22), the reviewer requested that we indicate the subcellular compartments, such as the cytoplasm and the nucleus. The reviewer also asked us to clarify which forms of PPAR-γ bind to DNA and to explain the meanings of the red square and the green and blue circles.

In response to the reviewer's comment, we have revised Figure 8 to provide clearer explanations and annotations. We intentionally focused on the events occurring in the cell nucleus, and therefore, did not explicitly indicate subcellular compartments. To address the reviewer's concerns, we have revised the figure caption to include essential information about the symbols used. Specifically, we have clarified that the blue circle represents a specific agonist of RXR-α that interacts with the RXR-α molecule (line 606), the green circle denotes an agonist that interacts with PPAR-γ (in part (a), lines 610-611), and the red square signifies an antagonist that interacts with the PPAR-γ molecule (in part (e), line 617). Additionally, we have provided information on the forms of PPAR-γ and their interactions with DNA, as suggested by the reviewer. We hope this revised figure and caption meet the reviewer's requirements and provide a clearer understanding of the events that we described.

COMMENT #5: Reviewer #3 also made a technical comment regarding the former Figure 7 (now revised as Figure 8, please see page 22 in the revised version of the manuscript). Specifically, the reviewer suggested that we replace "i" with "h" in the cartoon. Furthermore, the reviewer requested that we indicate the "closed" and "open" conformations of PPAR-γ.

We have made the necessary revisions to Figure 8. As requested by the reviewer, we have replaced "i" with "h" in the cartoon. Additionally, we have clearly labeled the "open" and "closed" conformations of PPAR-γ, providing further clarity on the structural dynamics of this protein. We have also taken this opportunity to specify which forms of PPAR-γ interact with DNA and to indicate the consequences of these interactions, thereby providing a more comprehensive understanding of the underlying mechanisms. We appreciate the reviewer's input and hope that these revisions have enhanced the figure's accuracy and usefulness.

COMMENT #6: Reviewer #3 also requested that we add PPAR-γ to the former Figure 8 (now revised as Figure 9, page 22 in the revised version of the manuscript).

In response to the reviewer's request, we have revised Figure 9 (formerly Figure 8) to include PPAR-γ-containing protein complexes (page 25 of the revised version). Specifically, we have added illustrations of the activatory complex, which induces gene expression, on the left side of the figure, and the repressory complex on the right side. To ensure consistency and clarity, we have also verified that the main text of the manuscript provides a clear explanation of the formation and function of both complexes, as depicted in the revised Figure 9. For convenience, the relevant explanations can be found on pages 23 (in the second and third paragraphs from the bottom) and page 21 (in the two paragraphs following the new Figure 7). We hope that this revised figure, along with the accompanying text, provides a comprehensive and accurate representation of the PPAR-γ-containing protein complexes and their roles in regulation of gene expression.

Thank you for considering our work.

Sincerely yours,
MEZENTSEV, Alexandre, PhD 
